# Representation Ensembling for Synergistic Lifelong Learning with Quasilinear Complexity

## Abstract

In lifelong learning, data are used to improve performance not only on the current task, but also on previously encountered, and as yet unencountered tasks. In contrast, classical machine learning, which we define as starting from a blank slate, or *tabula rasa* uses data only for the single task at hand. While typical transfer learning algorithms can improve performance on future tasks, their performance on prior tasks degrades upon learning new tasks (called forgetting). Many recent approaches for continual or lifelong learning have attempted to *maintain* performance on old tasks given new tasks. But striving to avoid forgetting sets the goal unnecessarily low. The goal of lifelong learning should be to not only improve performance on future tasks (forward transfer) but also to improve performance on past tasks (backward transfer) with any new data. Our key insight is that we can synergistically ensemble representations that were learned independently on disparate tasks to enable both forward and backward transfer. This generalizes ensembling decisions (like in decision forests) and complements ensembling dependently learned representations (like in multitask learning). Moreover, we ensemble representations in quasilinear space and time. We demonstrate this insight with two algorithms: representation ensembles of (1) trees and (2) networks. Both algorithms demonstrate forward and backward transfer in a variety of simulated and benchmark data scenarios, including tabular, image, and spoken, and adversarial tasks. This is in stark contrast to the reference algorithms we compared to, most of which failed to transfer either forward or backward, or both, despite that many of them require quadratic space or time complexity.

## 1 Introduction

Learning is the process by which an intelligent system improves performance on a given task by leveraging data (Mitchell, 1999). In classical machine learning, the system is often optimized for a single task (Vapnik & Chervonenkis, 1971; Valiant, 1984). While it is relatively easy to *simultaneously* optimize for multiple tasks (multi-task learning) (Caruana, 1997), it has proven much more difficult to *sequentially* optimize for multiple tasks (Thrun, 1996; Thrun & Pratt, 2012). Specifically, classical machine learning systems, and natural extensions thereof, exhibit "catastrophic forgetting" when trained sequentially, meaning their performance on the prior tasks drops precipitously upon training on new tasks (McCloskey & Cohen, 1989; McClelland et al., 1995). However, learning could be lifelong, with agents continually building on past knowledge and experiences, improving on many tasks given data associated with any task. For example, learning a second language often improves performance in an individual's native language (Zhao et al., 2016).

In the past 30 years, a number of sequential task learning algorithms have attempted to overcome catastrophic forgetting. These approaches naturally fall into one of two camps. In one, the algorithm has fixed resources, and so must reallocate resources (essentially compressing representations) in order to incorporate new knowledge Kirkpatrick et al. (2017); Zenke et al. (2017); Li & Hoiem (2017); Schwarz et al. (2018); Finn et al. (2019). Biologically, this corresponds to adulthood, where brains have a nearly fixed or decreasing number of cells and synapses. In the other, the algorithm adds (or builds) resources as new data arrive (essentially ensembling representations) (Ruvolo & Eaton, 2013; Rusu et al., 2016; Lee et al., 2019). Biologically, this corresponds to development, where brains grow by adding cells, synapses, etc. A close resemblance

to this resource growing approach can be found in Sodhani et al. (2020) where the model adaptively expands the capacity when the capacity of the model saturates.

Approaches from both camps demonstrate some degree of continual (or lifelong) learning (Parisi et al., 2019). In particular, they can sometimes learn new tasks while not catastrophically forgetting old tasks. However, as we will show, many state of the art lifelong learning algorithms are unable to transfer knowledge forward (to future unseen tasks) and most of them do not transfer backward (to previously seen tasks). With high enough sample sizes, some of them are able to transfer forward or backward, but transfer is more important in low sample size regimes (Chen & Liu, 2016; Lee et al., 2019). This inability to effectively transfer in low-sample size regimes has been identified as one of the key obstacles limiting the capabilities of artificial intelligence (Pearl, 2019; Marcus & Davis, 2019). Our work falls into the (arguably simpler) resource growing camp in which each new task is learned with additional representational capacity.

Prior work illustrates that ensembling learners can yield huge advantages in a wide range of applications. For example, in classical machine learning, ensembling trees leads to state-of-the-art random forest (Breiman, 2001) and gradient boosting tree algorithms (Chen & Guestrin, 2016). Similarly, ensembling networks shows promising results in various real-world applications (Qiu et al., 2014; Potes et al., 2016). Wang et al. (2003) used weighted ensemble of learners in a streaming setting with distribution shift. TrAdaBoost Dai et al. (2007), boosts ensembles to enable transfer learning. In continual learning scenarios, many algorithms have been built on these ideas by ensembling dependent representations. For example, Learn++ Polikar et al. (2001) boosts ensembles of weak learners learned over different data sequences in class incremental lifelong learning settings van de Ven et al. (2022). Model Zoo (Ramesh & Chaudhari, 2021), uses the same boosting approach in task incremental lifelong learning scenarios van de Ven et al. (2022).

Another group of algorithms, ProgNN (Rusu et al., 2016) and DF-CNN (Lee et al., 2019) learn a new column with each new task and ensembles the columns while inferring on a certain task. The only difference between ProgNN and DF-CNN is that ProgNN has forward connections to the current column from all the past columns. This creates the possibility of forward transfer while freezing backward transfer. However, the forward connections in ProgNN render it computationally inefficient for a large number of tasks. DF-CNN gets around this problem by learning a common knowledge base and thereby, creating the possibility of backward transfer.

Recently, many modular approaches have been proposed in the literature which propose an improvement on the process how ProgNN grows capacity with more tasks. These methods consider the capacity for each task being composed of modules that can be shared across tasks and grown as necessary. For example, PackNet Mallya & Lazebnik (2018) starts with a fixed capacity network and trains for additional task by freeing up portion of the network capacity using iterative pruning. Veniat et al. (2020) trains additional modules to the model as a new task is seen and the old modules are selectively used for the corresponding task. Ostapenko et al. (2021) improved the memory efficiency of the modular methods by adding new modules according to the complexity of the new tasks. Mehta et al. (2021) proposed non-parametric factorization of the layer weights that promotes sharing of the weights between tasks. However, all of modular methods described so far also lack backward transfer as the old modules are not updated with the new tasks. Dynamically Expandable Representation (DER) (Yan et al., 2021) proposed an improvement over the modular approaches where the model capacity is dynamically expanded and the model is fine-tuned by replaying a portion of the old task data along with the new task data. This approach achieves backward transfer between tasks as reported by the authors in the experiments.

Our key innovation is that one can ensemble independent representations, thereby benefiting from past representations without biasing future representations. Moreover, we introduce a channel layer to enable the representations to interact with one another, thereby enabling computationally efficient forward and backward transfer.

Specifically, we introduce two complementary lifelong learning algorithms, one based on ensembling decision forests (Syngeristic Forests, SynF), and another based on ensembling deep networks (Synergistic Networks, SynN). Both decision forests and deep networks learn a task representation in terms of polytopes that partition the feature space (Priebe et al., 2020). SynF and SynN ensemble sets of polytopes learned from each task by aggregating discriminative information across tasks via a channel layer. Additionally, we propose learning

metrics which are crucial in quantifying lifelong learning capabilities. We use simulation study to explore some key properties of our proposed algorithms. In our experiments, we consider a simplified multi-pass (offline) learning environment akin to those previously published (Kirkpatrick et al., 2017; Schwarz et al., 2018; Zenke et al., 2017; Li & Hoiem, 2017; Rusu et al., 2016; Lee et al., 2019), where we know the task identities during inference and the tasks are streaming but the data within the tasks are batched. Moreover, our approach achieves backward transfer by keeping all the training samples from the old tasks and update the model with additional tasks like the total replay approaches. We have done ablation experiment on vision data showing the impact of varying the amount of data replayed. The replay approaches (van de Ven et al., 2020; Robins, 1995; Shin et al., 2017; van de Ven et al., 2020) demonstrated the existence of algorithms that keep old data and mitigate catastrophic forgetting. However, previously proposed replay algorithms do not demonstrate backward transfer in our experiments. On the contrary, both SynF and SynN demonstrate forward and backward transfer, while maintaining computational efficiency in vision and language benchmark applications. Although the algorithms presented here are primarily resource building, we illustrate that they can effectively leverage prior representations to operate in resource constrained scenarios. This ability implies that the algorithm can convert from a "juvenile" resource building state to the "adult" resource recycling state – all while maintaining key lifelong learning capabilities and efficiencies.

## 2 Background

### 2.1 Classical Machine Learning

Classical supervised learning (Mohri et al., 2018) considers random variables $(X, Y) \sim \mathcal{D}$, where $X$ is an $\mathcal{X}$-valued input, $Y$ is a $\mathcal{Y}$-valued label (or response), and $\mathcal{D}$ is the joint distribution of $(X, Y)$. Given a loss function $\ell : \mathcal{Y} \times \mathcal{Y} \to [0, \infty)$, the goal is to find the hypothesis (also called predictor), $h : \mathcal{X} \to \mathcal{Y}$ that minimizes expected loss, or *risk*, $R(h) = \mathbb{E}_{(X,Y) \sim \mathcal{D}} [\ell(h(X), Y)]$. A learning algorithm is a function $f$ that maps data sets ($n$ training samples) to a hypothesis, where a data set $\mathbf{S}_n = \{X_i, Y_i\}_{i=1}^n$ is a set of $n$ input/response pairs. Assume $n$ samples of $(X, Y)$ pairs are independently and identically distributed from some true but unknown $\mathcal{D}$ (Mohri et al., 2018). A learning algorithm is evaluated on its generalization error (or expected risk): $\mathbb{E}[R(f(\mathbf{S}_n))] = \mathbb{E}\left[R(\hat{h}_n)\right]$, where the expectation is taken with respect to the true but unknown distribution governing the training data, $\mathcal{D}_n$. The goal is to choose a learner $f$ that learns a hypothesis $\hat{h}_n$ using $n$ training samples that has a small generalization error for the given task (Bickel & Doksum, 2015).

### 2.2 Lifelong Learning

Lifelong supervised learning generalizes classical supervised machine learning in a few ways: (i) instead of one task, there is an environment $\mathcal{T}$ of (possibly infinitely) many tasks, (ii) evaluation data-label pair $(X, Y)$ for each task sampled from some distribution $\mathcal{D}$ arrive sequentially, rather than in batch mode, and (iii) there are computational complexity constraints on the learning algorithm and hypotheses. In supervised learning settings, one can consider the following risk for a particular task $t$ with n random training samples $\mathbf{S}_n$ distributed as $\mathcal{D}_n$:

$$R^t(f(\mathbf{S}_n)) = R^t(\hat{h}_n) = \mathbb{E}_{(X,Y) \sim \mathcal{D}}[\ell(\hat{h}_n(X), Y)]. \tag{1}$$

Note that the data $\mathbf{S}_n$ is a random variable and it may contain data related to any number of tasks (potentially all the tasks) in the environment. One may take expectation with respect to $\mathcal{D}_n$ for averaging out the randomness in the risk due to $\mathbf{S}_n$ and consider the generalization error for the task as:

$$\mathcal{E}_f^t(\mathbf{S}_n) = \mathbb{E}_{\mathbf{S}_n \sim \mathcal{D}_n}[R^t(f(\mathbf{S}_n))]. \tag{2}$$

We are given the error $\mathcal{E}_f^t(\mathbf{S}_n)$ for $t = 1, \cdots, T$ and a weight for each task $m_t$ corresponding to the extent the learner prioritizes task $t$ such that $\sum_{t=1}^T m_t = 1$ and $m_t \geq 0$. For example, consider a biological learner which may prioritize the tasks necessary for its survival more than other tasks. Letting $\mathcal{E}_f^{\mathcal{T}}(\mathbf{S}_n) = \sum_{t \in \mathcal{T}} m_t \, \mathcal{E}_f^t(\mathbf{S}_n)$

and given a class of learners $\mathcal{F}$, the goal of a multi-task learner is to find an $f \in \mathcal{F}$ such that:

$$\begin{array}{ll} \text{minimize} & \mathcal{E}_f^{\mathcal{T}}(\mathbf{S}_n) \\ \text{subject to} & f \in \mathcal{F} \end{array} . \tag{3}$$

A multi-task learner has access to all the data at the same time. However, a lifelong learning agent has sequential access to the data and the data sequence is updated with each encounter of $m$ new training sample point as $\mathbf{S}_{n-m} \to \mathbf{S}_n$. The lifelong learning algorithm $f$ takes the new training sequence $\mathbf{S}_n$ and previous hypothesis $\hat{h}_{n-m}$ as input and outputs a new hypothesis $h_n$: $f(\mathbf{S}_n, \hat{h}_{n-m}) \to \hat{h}_n$. Now, given $n$ samples at any time instant, the goal of a lifelong learner is to solve Equation 3. Implicit in Equation 3 is that we are not only concerned not just with past tasks, but also all possible future tasks. Moreover, Equation 3 allows to prioritize some tasks more than other tasks. For example, biological agents may prioritize the tasks crucial for its survival. However, we use equal priority for all the tasks in this paper which is a common practice in the literature Rusu et al. (2016); Lee et al. (2019); Ramesh & Chaudhari (2021). That said, we are not explicitly solving Objective equation 3 in our proposed approach.

The computational complexity constraints for lifelong learning are crucial, though often implicit. For example, consider the algorithm that stores all the data, and then retrains everything from scratch each time a new sample arrives. Without computational constraints, such an algorithm could be classified as a lifelong learner; we do not think such a label is appropriate for that algorithm. Thus, we only consider learners $f$ lifelong learners assuming their performance scales sub-quadratically with sample size (see below for details). The goal in lifelong learning therefore is, given new data and a new task, use all the existing data to achieve lower generalization error on this new task, while also using the new data to obtain a lower generalization error on the previous tasks. This is distinct from classical online learning scenarios (Cesa-Bianchi & Lugosi, 2006), because the previously experienced tasks may recur, so we are concerned about maintaining and improving performance on those tasks as well. In "task-aware" scenarios, the learner is aware of all task details for all tasks, meaning that the hypotheses are of the form $h : \mathcal{X} \times \mathcal{T} \to \mathcal{Y}$. In "task-unaware" (or agnostic (Zeno et al., 2018)) scenarios the learner may not know that the task has changed at all, which means that the hypotheses are of the form $h : \mathcal{X} \to \mathcal{Y}$. We only address task-aware scenarios here.

## 2.3 Reference algorithms

We compared our approaches to 14 reference lifelong learning methods. These algorithms can be classified into two groups based on whether they add capacity resources per task, or not. Among them, ProgNN (Rusu et al., 2016) and Deconvolution-Factorized CNNs (`DF-CNN`) (Lee et al., 2019) learn new tasks by building new resources. For ProgNN, for each new task a new "column" of network is introduced. In addition to introducing this column, lateral connections from all previous columns to the new column are added. These lateral connections are computationally costly, as explained in Subsection 4.3. `DF-CNN` (Lee et al., 2019) is a lifelong learning algorithm that improves upon ProgNN by introducing a knowledge base with lateral connections to each new column, thereby avoiding all pairwise connections, and dramatically reducing computational costs. We also compare two variants of exact replay (Total Replay and Partial Replay) using the code provided by van de Ven et al. (2020). According to the code, both Total and Partial Replay store all the data they have ever seen, but Total Replay replays all of it upon acquiring a new task, whereas Partial Replay replays $M$ samples, randomly sampled from the entire corpus, whenever we acquire a new task with $M$ samples. We have also compared our appraoch with more constrained ways of replaying old task data like- Model Zoo (Ramesh & Chaudhari, 2021), Averaged Gradient Episodic Memory (`A-GEM`) (Chaudhry et al., 2018), Experience Replay (`ER`) (Chaudhry et al., 2019) and Task-based Accumulated Gradients (`TAG`) (Malviya et al., 2021) for lifelong learning. Among them Model Zoo ensembles multiple representations using the boosting approach. In Model Zoo, the total number of models within the ensemble (the number of episodes) was capped at the total number of tasks to make it comparable with our approach. For `A-GEM` and `ER`, the size of episodic memory is set to store 1 example per class. On the other hand, TAG stores the gradients or directions the model took while learning a specific task instead of storing past examples.

The other five algorithms, Elastic Weight Consolidation (`EWC`) (Kirkpatrick et al., 2017), Online-EWC (`O-EWC`) (Schwarz et al., 2018), Synaptic Intelligence (`SI`) (Zenke et al., 2017), Learning without Forgetting (`LwF`) (Li & Hoiem, 2017), and "None," all have fixed capacity resources. For the baseline "None", the

network was incrementally trained on all tasks in the standard way while always only using the data from the current task. The implementations for all of the algorithms are adapted from open source codes (Lee et al., 2019; van de Ven & Tolias, 2019); for implementation details, see Appendix D.

## 3 Evaluation Criteria

Others have previously introduced criteria to evaluate transfer, including forward and backward transfer (Lopez-Paz & Ranzato, 2017; Benavides-Prado et al., 2018; Díaz-Rodríguez et al., 2018; Veniat et al., 2020). These definitions typically compare the difference, rather than the ratio, between learning with and without transfer. Pearl Judea (2018) introduced the transfer benefit ratio, which builds directly off relative efficiency from classical statistics (Bickel & Doksum, 2015). Our definitions are closely related to Pearl's definition. Below, we introduce new metrics for lifelong learning (§3.1), describe our desiderata (§3.2) for a lifelong learning metrics which are satisfied by our proposed metrics, and illustrate why the desiderata are useful in a concrete example (§3.3).

### 3.1 Learning Efficiency

*Learning efficiency* is the ratio of the generalization error of an algorithm that has learned on one dataset, as compared to the generalization error of that same algorithm on a different dataset. Typically, we are interested in situations where the former dataset is a subset of the latter dataset. Consider a lifelong learning environment with total $T$ tasks introduced to the learning agent sequentially. Let $R^t$ be the risk associated with Task $t$, and $\mathbf{S}^i$ be the data that is specifically associated with any Task $i$ with sample size $n_i$, so $R^t(f(\mathbf{S}^t))$ is the risk on Task $t$ of the hypothesis learned by $f$ only using Task $t$ data, and $R^t(f(\bigcup_{i=1}^{T} \mathbf{S}^i))$ denotes the risk on Task $t$ of the hypothesis learned on all the data up to Task $T$. Note that, $\sum_{i=1}^{T} n_i = n$.

**Definition 1 (Learning Efficiency)** *The learning efficiency of algorithm f for given Task t with total sample size n is:*

$$\mathsf{LE}_n^t(f) := \frac{\mathcal{E}_f^t(\mathbf{S}^t)}{\mathcal{E}_f^t(\bigcup_{i=1}^{T} \mathbf{S}^i)}. \tag{4}$$

*We say that algorithm f has transferred across all the tasks up to T with data $\mathbf{S}$ if and only if $\mathsf{LE}_n^t(f) > 1$ for all the tasks up to T.*

To evaluate a lifelong learning algorithm while respecting the streaming nature of the tasks, it is convenient to consider two extensions of learning efficiency. *Forward* learning efficiency is the expected ratio of the generalization error of the learning algorithm with (i) access only to Task $t$ data, to (ii) access to the data up to and including the last observation from Task $t$. This quantity measures the relative effect of previously seen out-of-task data on the performance on Task $t$.

**Definition 2 (Forward Learning Efficiency)** *The forward learning efficiency of f for task t given n samples is :*

$$\mathsf{FLE}_n^t(f) := \frac{\mathcal{E}_f^t(\mathbf{S}^t)}{\mathcal{E}_f^t(\bigcup_{i=1}^{t} \mathbf{S}^i)}. \tag{5}$$

We say an algorithm (positively) forward transfers for task $t$ if and only if $\mathsf{FLE}_n^t(f) > 1$. In other words, if $\mathsf{FLE}_n^t(f) > 1$, then the algorithm has used data associated with past tasks to improve performance on task $t$. Note that a learner has only forward transfer from the past tasks to a specific task only when the task is introduced to the learner.

One can also determine the rate of *backward* transfer by comparing the generalization error $\mathcal{E}_f^t(\bigcup_{i=1}^{t} \mathbf{S}^i)$ to the generalization error of the hypothesis learned having seen the entire training dataset up to Task $T$. More formally, backward learning efficiency is the ratio of the generalization error of the learned hypothesis with (i) access to the data up to and including the last observation from task $t$, to (ii) access to the entire dataset. Thus, this quantity measures the relative effect of future task data on the performance on Task $t$.

**Definition 3 (Backward Learning Efficiency)** *The backward learning efficiency of f for Task t given n samples is*

$$\mathsf{BLE}_n^t(f) := \frac{\mathcal{E}_f^t(\bigcup_{i=1}^t \mathbf{S}^i)}{\mathcal{E}_f^t(\bigcup_{i=1}^T \mathbf{S}^i)}. \tag{6}$$

We say an algorithm (positively) backward transfers to Task $t$ from all the tasks $T$ if and only if $\mathsf{BLE}_n^t(f) > 1$. We can report $\mathsf{BLE}_n^t(f)$ for each $t$ as we gradually increase the number of total task $T$ in the environment or we can report the final $\mathsf{BLE}_n^t(f)$ for each $t$ after we are done adding task to the environment as a summary. The former measure shows the dynamics of the task specific performance whereas the latter one shows an average performance from all the tasks. In summary, if $\mathsf{BLE}_n^t(f) > 1$, then the algorithm has used data associated with future tasks to improve performance on past tasks.

After observing $T$ tasks, the extent to which the LE for the $i^{th}$ task comes from forward transfer versus from backward transfer depends on the order of the tasks. If we have a sequence in which tasks do not repeat, learning efficiency for the first task is all backward transfer, for the last task it is all forward transfer, and for the middle tasks it is a combination of the two. In general, LE factorizes into FLE and BLE:

$$\mathsf{LE}_n^t(f) = \frac{\mathcal{E}_f^t(\mathbf{S}^t)}{\mathcal{E}_f^t(\bigcup_{i=1}^T \mathbf{S}^i)} = \frac{\mathcal{E}_f^t(\mathbf{S}^t)}{\mathcal{E}_f^t(\bigcup_{i=1}^t \mathbf{S}^i)} \times \frac{\mathcal{E}_f^t(\bigcup_{i=1}^t \mathbf{S}^i)}{\mathcal{E}_f^t(\bigcup_{i=1}^T \mathbf{S}^i)}. \tag{7}$$

Throughout, we will report log LE so that positive learning corresponds to LE $> 1$. In a lifelong learning environment having $T$ tasks drawn with replacement from $\mathcal{T}$, learner $f$ $\bm{m}$-lifelong learns tasks $t \in \mathcal{T}$ if the log of the convex combination of learning efficiencies is greater than 0, that is,

$$\log \sum_{t \in \mathcal{T}} m_t \cdot \mathsf{LE}_n^t(f) > 0 \tag{8}$$

where $m_t$ corresponds to the extent to which the learner prioritizes a certain task $t$. Note that when $m_t$ is equal for each task, the learner has to excel equally in each task. We say an agent has **synergistically learned** in an environment of $T$ tasks if the agent has positively learned, i.e., the quantity in equation 8 is positive for all of the possible convex combinations of all the tasks up to $T$.

### 3.2 Desiderata

In this section, we explore the efficacy of our proposed metrics in (§3.1) for quantifying lifelong learning capability of an agent. While there is no universal set of desiderata, here we list our desiderata, and explain why we believe these are useful for deeply investigating and comparing the performance of lifelong learning machines. Specifically, we desire the following for our evaluation criteria.

1. Explicitly quantify **transfer**, as opposed to accuracy. It is possible for accuracy for a given learner to increase with more tasks without transfer (for example, if the subsequent tasks happen to get easier). This phenomenon is highlighted again in (§6.1.4). Definition equation 4 satisfies this desiderata by normalizing performance appropriately.

2. A general definition of learning efficiency which we can apply in multiple distinct scenarios, including generic transfer, and also both forward and backward transfer. This is because both forward and backward transfer are special cases of transfer learning achieved from two different streams of data, i.e., past task data and future task data, respectively. This property simplifies the need to derive multiple distinct metrics which are difficult to compare to one another. Definitions equation 4, equation 5, and equation 6 use the same function, just with different data streams in the numerator and the denominator.

3. The total amount of transfer naturally decomposes in forward and backward transfer. This simplifies understanding the contribution of various aspects of transfer. Equation equation 7 illustrates that our definitions satisfy this desiderata. That being said there are other metrics in the literature

(Veniat et al., 2020) which could also be decomposed into forward and backward transfer instead of reporting an average performance if desired so.

4. Our criteria normalize accuracy. When comparing multiple approaches with high accuracy, small absolute gains translate into large relative gains. This is important because in general, once we get to high accuracy levels (e.g., 98% or so), we care deeply about gains in *relative* performance, that is, reducing error from 2% to 1% is a big deal. In contrast, if one reduces error from 49% to 48%, that is relatively less interesting and impactful. In both of the cases, the change in accuracy is 1% whereas learning efficiencies (LE) are 2 and 1.02, respectively. This desiderata is also common in statistical evaluations. For example, the classical approach for comparing two different estimators is the relative efficiency Bickel J & Doksum A (2015). That being said, the log of our criteria looks at absolute differences.

However, there is a downside associated with the fourth desideratum which may inflate or deflate the overall transfer due to noise. Note that in practice we calculate the empirical error and empirical risk in Equation 2 over several Monte Carlo repetitions. If the experiments are not repeated for enough repetitions, the calculated empirical errors will be noisy and this noise may be amplified through our proposed metrics as we are taking ratios of empirical errors.

### 3.3 A concrete example on lifelong learning metrics

Table 1: Learning metrics summarized on a hypothetical scenario. The metrics are color-coded as red if it improves, blue if it degrades and black if it is not impacted with additional data. Note that $\log \mathsf{LE}_n^t = \log \mathsf{FLE}_n^t + \log \mathsf{BLE}_n^t$.

| Metrics | | $R^1(f(S))$ | $R^2(f(S))$ | | $\log \mathsf{FLE}_n^1(f)$ | $\log \mathsf{FLE}_n^2(f)$ | FWT |
|---|---|---|---|---|---|---|---|
| Dataset | $S = S^1$ | 0.30 | 0.50 | | 0 | - | 0.50 |
| | $S = S^1 \bigcup S^2$ | 0.32 | 0.45 | | 0 | $-0.05$ | 0.50 |

| Metrics | | $\log \mathsf{LE}_n^1(f)$ | $\log \mathsf{LE}_n^2(f)$ | | $\log \mathsf{BLE}_n^1(f)$ | $\log \mathsf{BLE}_n^2(f)$ | BWT |
|---|---|---|---|---|---|---|---|
| Dataset | $S = S^1$ | 0 | - | | 0 | - | 0 |
| | $S = S^1 \bigcup S^2$ | $-0.03$ | $-0.05$ | | $-0.03$ | 0 | $-0.02$ |

The following example justifies how our proposed metrics satisfy all the four aforementioned desiderata. We compare and contrast our proposed metrics with the metrics proposed in Díaz-Rodríguez et al. (2018) on a hypothetical scenario, including accuracy, backward transfer (BWT), and forward transfer (FWT). Consider a lifelong learning environment with two tasks each having two classes. The tasks are introduced sequentially with $n_1$ samples from Task 1 and then $n_2$ samples from Task 2. The agent has a generalization error of $R^1(f(S^1)) = 0.3$ on Task 1 while it has access to the Task 1 dataset only, and a generalization error of $R^2(f(S^2)) = 0.4$ on Task 2, while it has access to the Task 2 dataset only. Now consider the scenario when the agent has the same hyper-parameters and sequential access to all the task datasets. Suppose the model has the generalization error on two tasks enumerated as in table 1. Note that the FLEs are given by: $\mathsf{FLE}_n^1(f) = \frac{R^1(f(\mathbf{S}^1))}{R^1(f(\mathbf{S}^1))} = 1$ and $\mathsf{FLE}_n^2(f) = \frac{R^2(f(\mathbf{S}^2))}{R^2(f(\mathbf{S}^1 \bigcup \mathbf{S}^2))} = 0.89$. The performance metrics can be summarized as in table 1.

As evident in Table 1, the transfer learning for Task 1 comes from backward transfer from Task 2 whereas for Task 2 it comes from forward learning from Task 1. As a summary, one can look at the final LEs over all the tasks after all tasks have been introduced. Note that in Table 1 the learning efficiencies are never greater than 1. However, the average accuracy on all tasks increased from 60% to 62%. Therefore, only using multi-task accuracy may falsely detect positive transfer. In Table 1, BWT can correctly identify an overall negative backward transfer or forgetting. However, being an average quantity, it can not resolve the overall backward transfer into individual task. As a result, a task with extremely high backward transfer may mask all the negative backward transfer from the other tasks giving a net positive transfer. On the other hand, FWT being at chance level fails to report any change in forward learnability with additional data.

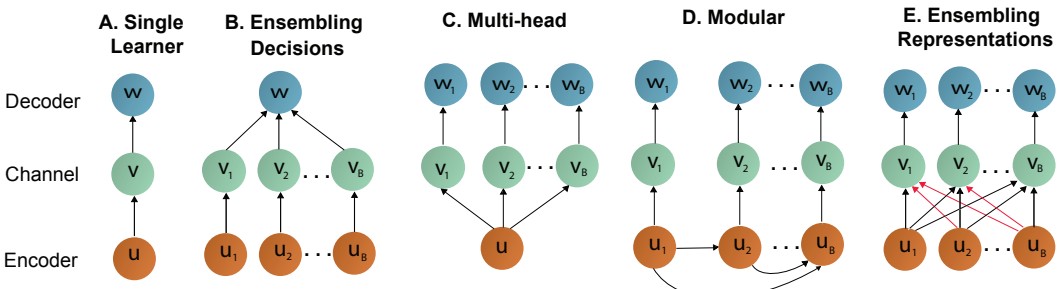

Figure 1: Schemas of composable hypotheses. A. Single task learner. B. Ensembling decisions (as output by the channels) is a well-established practice, including random forests and gradient boosted trees. C. Learning a joint representation or D. Ensembling representations (learned by the encoders) was previously used in lifelong learning scenarios, but were not trained independently as in E, thereby causing interference or forgetting. Note that the new encoders interact with the previous encoders through the channel layer (indicated by red arrows), thereby, enabling backward transfer. Again the old encoders interact with the future encoders (indicated by black arrows), thereby, enabling forward transfer.

## 4    Representation ensembling algorithms

In this section, we provide an abstract idea of our approach and we refine the details of the algorithms further in Subsection 4.1 and 4.2. We start by decomposing a learner into three components: an encoder, a channel, and a decoder (Cover & Thomas, 2012; Cho et al., 2014): $h(\cdot) = w \circ v \circ u(\cdot)$. Figure 1 shows these three components as the building blocks of different learning schemas. The encoder, $u : \mathcal{X} \mapsto \tilde{\mathcal{X}}$, maps an $\mathcal{X}$-valued input into an internal representation space $\tilde{\mathcal{X}}$ (Vaswani et al., 2017; Devlin et al., 2018). The channel $v : \tilde{\mathcal{X}} \mapsto \Delta_{\mathcal{Y}}$ maps the transformed data into a posterior distribution (or, more generally, a score). Finally, a decoder $w : \Delta_{\mathcal{Y}} \mapsto \mathcal{Y}$, produces a predicted label.

For example, in Figure 1A, consider we have a dataset partitioned into a training and a held-out set. Now we can learn a decision tree using the training data which will give us the encoder. Next, by pushing the held-out dataset through the tree, we can learn the channel, i.e., posteriors in the leaf-nodes. The channel thus gives scores for each data point denoting the probability of that data point belonging to a specific class. Using separate sets of data to learn the encoder and the channel results in less bias in the estimated posterior in the channels as in 'honest trees' (Breiman et al., 1984; Denil et al., 2014; Athey et al., 2019). Finally, the decoder provides the predicted class label using arg max over the posteriors from the channel. See Appendix A for a more detailed and concrete example using a decision tree.

According to Figure 1B, one can generalize the above decomposition by allowing for multiple encoders. Given $B$ different encoders, one can attach a single channel to each encoder, yielding $B$ different channels. Doing so requires generalizing the definition of a decoder so that it would operate on multiple channels. Such a decoder ensembles the *decisions*, because here each channel provides the final output based on the encoder. This is the learning paradigm behind boosting (Freund, 1995) and bagging (Breiman, 1996)—indeed, decision forests are a canonical example of a decision function operating on a collection of $B$ outputs (Breiman, 2001). A decision forest learns $B$ different decision trees, each of which has a tree structure corresponding to an encoder. Each tree is assigned a channel that outputs each tree's vote that an observation is in any class. The decoder outputs the most likely class averaged over the trees.

Although the task specific structure in Figure 1B can provide useful decision on the corresponding task, they can not, in general, provide meaningful decisions on other tasks because those tasks might have completely different class labels. Therefore, in the multi-head structure (Figure 1C) a single encoder is used to learn a joint representation from all the tasks and a separate channel is learned for each task to get the score or class conditional posteriors for each task which is followed by each task specific decider (Kirkpatrick et al., 2017; Schwarz et al., 2018; Zenke et al., 2017). Further modification of the multi-head structure allows ProgNN or other modular approaches to learn separate encoder for each task with forward connections from the past

encoders to the current one (Figure 1D). This creates the possibility of having forward transfer while freezing backward transfer. Note that if the encoders are learned independently across different tasks, they may have learned useful *representations* that the tasks can mutually leverage.

Our approach is based on the learning scheme in Figure 1E. This scheme requires generalizing the definition of a channel so that it can operate on multiple encoders. The result is that the channels **ensemble representations** (learned by the encoders), rather than decisions (learned by the channels) as in Figure 1B. In this scenario, like with bagging and boosting, the ensemble of channels then feeds into the single task specific decoder. When each encoder has learned complementary representations, this representation ensembling approach has certain appealing properties, particularly in multiple task scenarios, including lifelong learning.

We have developed two different representation ensembling algorithms based on bagging of the encoders which are trained on different tasks. In either of the algorithms, as new data from a new task arrives, our algorithm first builds a new independent encoder. Then, it builds the channel for this new task by pushing the out-of-bag or held-out data (data not seen by the encoder while training) through the existing encoders. Thus the channel integrates information across all existing encoders using the new task data, thereby enabling forward transfer. At the same time, it can push old task data through the new encoders to update the channels from the old tasks, thereby enabling backward transfer. In either case, new test data are passed through all existing encoders and corresponding channels to make a prediction. As we will show empirically, these two ensemble methods achieve backward transfer on several benchmark datasets which is a desirable property for lifelong learners. It is shown in Wyner et al. (2017) that both bagging and boosting asymptotically converge to the Bayes optimal solution. However, for finite sample size and similar model complexity, we empirically find bagging approach to lifelong learning performs better than that of boosting when the training sample size is low (see Figure 3) whereas boosting performs better on large training sample size (See main text Figure 7, 8 and Appendix Figure 4).

The key to both of our algorithms is the realization that both forests and networks partition feature space into a union of polytopes (Priebe et al., 2020). Thus, the internal representation learned by each can be considered a sparse vector encoding which polytope a given sample resides in. We can combine the discriminative information over different sets of polytopes learned over different tasks by populating the polytopes with the corresponding task data and thereby, learn a channel for that specific task (see Appendix A, B and C for detailed description of the proposed approach).

### 4.1 Synergistic Forests

Synergistic Forests (SYNF) ensemble decision trees or forests. For each task, the encoder $u_t$ of a SYNF is the representation learned by a decision forest (Amit & Geman, 1997; Breiman, 2001). The leaf nodes of each task-corresponding decision forest partition the input space $\mathcal{X}$ into polytopes (Breiman et al., 1984). The channel then learns the class-conditional posteriors by populating the polytopes with out-of-bag samples and taking class votes, as in "honest trees" (Breiman et al., 1984; Denil et al., 2014; Athey et al., 2019). Each channel outputs the posteriors averaged across the collection of forests learned over different tasks. The decoder $w_t$ outputs the argmax to produce a single prediction. Recall that honest decision forests are universally consistent classifiers and regressors (Athey et al., 2019), meaning that with sufficiently large sample sizes, under suitable though general assumptions, they will converge to minimum risk. Thus, the single task version of this approach simplifies to an approach called "Uncertainty Forests" (Mehta et al., 2019). Table 1 in the appendix lists the hyperparameters used in the CIFAR experiments.

### 4.2 Synergistic Networks

A Synergistic Network (SYNN) ensembles deep networks. For each task, the encoder $u_t$ in an SYNN is the "backbone" of a deep network (DN), including all but the final layer. Thus, each $u_t$ maps an element of $\mathcal{X}$ to an element of $\mathbb{R}^d$, where $d$ is the number of neurons in the penultimate layer of the DN. The channels are learned via $k$-Nearest Neighbors ($k$-NN) (Stone, 1977) over the $d$ dimensional representations of $\mathcal{X}$. Recall that a $k$-NN, with $k$ chosen such that as the number of samples goes to infinity, $k$ also goes to infinity,

while $\frac{k}{n} \to 0$, is a universally consistent classifier (Stone, 1977). We use $k = 16 \log_2 n$, which satisfies these conditions.The decoder is the same as above.

SynN differs from ProgNN in two key ways. First, recall that ProgNN builds a new neural network "column" for each new task, and also builds lateral connections between the new column and all previous columns. *In contrast,* SynN *excludes those lateral connections, thereby greatly reducing the number of parameters and train time.* Moreover, this makes each representation independent, thereby potentially avoiding interference across representations. Second, for inference on task $j$ data, assuming we have observed tasks up to $J > j$, ProgNN only leverages representations learned from tasks up to $j$, thereby excluding tasks $j + 1, \ldots, J$. *In contrast,* SynN *leverages representations from all $J$ tasks, a key difference which enables backward transfer.* SynF adds yet another difference as compared to SynN by replacing the deep network encoders with random forest encoders. This has the effect of making the capacity, space complexity, and time complexity scale with the complexity and sample size of each task. In contrast, both ProgNN and SynN have a fixed capacity for each task, even if the tasks have very different sample sizes and complexities.

### 4.3 A computational taxonomy of lifelong learning

Lifelong learning approaches can be divided into those with fixed computational space resources, and those with growing space resources. We, therefore, quantify the computational space and time complexities of the internal representation of a number of algorithms. The space complexity of the learner refers to the amount of memory space needed as a function of input sample size to train the learner (Kuo & Zuo, 2003). We also study the representation capacity of these algorithms. Capacity is defined as the size of the subset of models that is achievable by the learning algorithm (Zhang et al., 2021).

We use the soft-O notation $\tilde{\mathcal{O}}$ to quantify complexity (van Rooij et al., 2019). Letting $n$ be the sample size and $T$ be the number of tasks, we write that the capacity, space or time complexity of a lifelong learning algorithm is $f(n, t) = \tilde{\mathcal{O}}(g(n, T))$ when $|f|$ is bounded above asymptotically by a function $g$ of $n$ and $T$ up to a constant factor and polylogarithmic terms. For simplifying the calculation, we make the following assumptions:

1. Each task has the same number of training samples.

2. Capacity grows linearly with the number of trainable parameters in the model.

3. The number of epochs is fixed for each task and is not arbitrarily large in comparison with the total sample size $n$.

4. For the algorithms with dynamically expanding capacity, we assume the worst case scenario where an equal amount of capacity is added to the model with an additional task.

Assumption 3 enables us to write time complexity as a function of the sample size. Table 2 summarizes the capacity, space and time complexity of several reference algorithms, as well as our SynN and SynF. For space and time complexity, the table shows results as a function of $n$ and $T$, as well as the common scenario where sample size per task is fixed and therefore proportional to the number of tasks, $n \propto T$. For detailed calculation of time complexity see Appendix E.

Parametric lifelong learning methods have a representational capacity which is invariant to sample size and task number. Although the space complexity of some of these algorithms grow (because the size of the constraints grows, or they continue to store more and more data), their capacity is fixed. Thus, given a sufficiently large number of tasks, without placing constraints on the relationship between the tasks, eventually all parametric methods will catastrophically forget. EWC (Kirkpatrick et al., 2017), ONLINE EWC (Schwarz et al., 2018), SI (Zenke et al., 2017), and LwF (Li & Hoiem, 2017) are all examples of parametric lifelong learning algorithms.

Semi-parametric algorithms' representational capacity grows slower than sample size. For example, if $T$ is increasing slower than $n$ (e.g., $T \propto \log n$), then algorithms whose capacity is proportional to $T$ are semi-parametric. ProgNN (Rusu et al., 2016) is semi-parametric, nonetheless, its space complexity $\tilde{\mathcal{O}}(T^2)$ due to

Table 2: Capacity, space, and time constraints of the representation learned by various lifelong learning algorithms. We show soft-O notation ($\tilde{\mathcal{O}}(\cdot, \cdot)$ defined in main text) as a function of $n = \sum_t^T n_t$ and $T$, as well as the common setting where $n$ is proportional to $T$. Our algorithms and DF-CNN are the only algorithms whose space and time both grow quasilinearly with capacity growing.

| Parametric | Capacity | Space | | Time | | Examples |
|---|---|---|---|---|---|---|
| | $(n, T)$ | $(n, T)$ | $(n \propto T)$ | $(n, T)$ | $(n \propto T)$ | |
| parametric | 1 | 1 | 1 | $n$ | $n$ | `O-EWC`, `SI`, `LwF` |
| parametric | 1 | $T$ | $n$ | $nT$ | $n^2$ | `EWC` |
| parametric | 1 | $n$ | $n$ | $nT$ | $n^2$ | Total Replay |
| semi-parametric | $T$ | $T^2$ | $n^2$ | $nT$ | $n^2$ | ProgNN |
| semi-parametric | $T$ | $T$ | $n$ | $n$ | $n$ | `DF-CNN` |
| semi-parametric | $T$ | $T+n$ | $n$ | $n$ | $n$ | SynN, Model Zoo, DER |
| non-parametric | $n$ | $n$ | $n$ | $n$ | $n$ | SynF, IBP-WF |

the lateral connections. Moreover, the time complexity for ProgNN also scales quadratically with $n$ when $n \propto T$. Thus, an algorithm that literally stores all the data it has ever seen, and retrains a fixed size network on all those data with the arrival of each new task, would have smaller space complexity and the same time complexity as ProgNN. For comparison, we implement such an algorithm and refer to it as Total Replay. `DF-CNN` (Lee et al., 2019) improves upon ProgNN by introducing a "knowledge base" with lateral connections to each new column, thereby avoiding all pairwise connections. Because these semi-parametric methods have a fixed representational capacity per task, they will either lack the representation capacity to perform well given sufficiently complex tasks, and/or will waste resources for very simple tasks. SynN and SynF eliminate the lateral connections between columns of the network, thereby reducing space complexity down to $\tilde{\mathcal{O}}(T)$. They store all the data to enable backward transfer, but retains linear time complexity. Because the time required for pushing the old task data though the old encoders and learning or updating channels is negligible in comparison with the time required for training a new encoder.

SynF is a non-parametric lifelong learning algorithm with its capacity, space and time complexity all as $\tilde{\mathcal{O}}(n)$, meaning that its representational capacity naturally increases with the complexity of each task. Apart from SynF, Indian Buffet Process for Weight Factors (`IBP-WF`) (Mehta et al., 2021) proposed the only other non-parametric lifelong learning algorithm to our knowledge.

## 5   Providing intuition of synergistic learning through simulations

In this section, we explore how relative position of the decision boundaries between two classes in two tasks can affect our proposed approach using simple toy simulations. For simulation study, we have used a deep network (`DN`) architecture with two hidden layers each having 10 nodes.

### 5.1   Synergistic learning in a simple environment

Consider a very simple two-task environment: Gaussian XOR and Gaussian Exclusive NOR (XNOR) (Figure 2A, see Appendix F for details). The two tasks share the exact same discriminant boundaries: the coordinate axes. Thus, transferring from one task to the other merely requires learning a bit flip of the class labels. We sample a total 750 samples from XOR, followed by another 750 samples from XNOR.

SynF and random forests (`RF`) achieve the same generalization error on XOR when training with XOR data (Figure 2Bi). But because `RF` does not account for a change in task, when XNOR data appear, `RF` performance on XOR deteriorates (it catastrophically forgets). In contrast, SynF continues to improve on XOR given XNOR data, demonstrating backward transfer. Now consider the generalization error on *XNOR* (Figure 2Bii). Both SynF and `RF` are at chance levels for XNOR when only XOR data are available. When XNOR data are available, `RF` must unlearn everything it learned from the XOR data, and thus its performance on XNOR starts out nearly maximally inaccurate, and quickly improves. On the other hand,

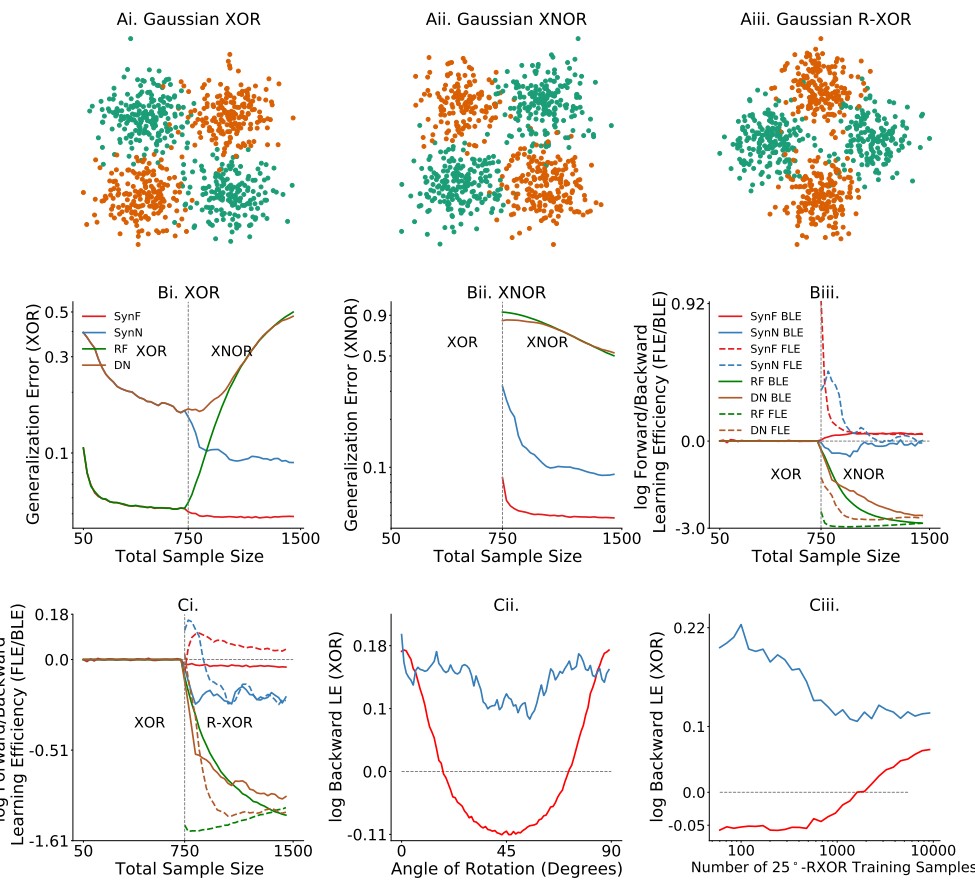

Figure 2: **Synergistic Forest and Synergistic Network demonstrate forward and backward transfer.** The learner is trained from scratch for each sample size so that we can observe the impact of increasing sample size on our algorithms. (*A*) 750 samples from: (*Ai*) Gaussian XOR, (*Aii*) XNOR, which has the same optimal discriminant boundary as XOR, and (*Aiii*) R-XOR, which has a discriminant boundary that is uninformative, and therefore adversarial, to XOR. (*Bi*) Generalization error for XOR, and (*Bii*) XNOR of both SynF (red), RF (green), SynN(blue), DN(dark orange). SynF outperforms RF on XOR when XNOR data is available, and on XNOR when XOR data are available. The same result is true for SynN sand DN. (*Biii*) Forward and backward learning efficiency of SynF are positive for all sample sizes, and are negative for all sample sizes for RF. Again, FLE and BLE is higher for SynNcompared to those of DN. (*Ci*) In an adversarial task setting (100 samples of XOR followed by 100 samples of R-XOR), SynFand SynN gracefully forgets XOR, whereas RFand DN demonstrate catastrophic forgetting and interference. (*Cii*) log BLE with respect to XOR is positive when the optimal decision boundary of $\theta$-XOR is similar to that of XOR (e.g. angles near $0°$ and $90°$), and negative when the discriminant boundary is uninformative, and therefore adversarial, to XOR (e.g. angles near $45°$). (*Ciii*) BLE is a nonlinear function of the source training sample size (XOR sample size is fixed at 500). For SynN experiments we did 100 repetitions and reported the results after smoothing it using moving average with a window size of 5. For the SynF experiments we used 1000 repetitions and reported the mean of these repetitions.

because SynF can leverage the encoder learned using the XOR data, upon getting *any* XNOR data, it immediately performs quite well, and then continues to improve with further XNOR data, demonstrating forward transfer (Figure 2Biii). SynF demonstrates positive forward and backward transfer for all sample sizes, whereas RF fails to demonstrate forward or backward transfer, and eventually catastrophically forgets the previous tasks. Qualitatively similar results are visible for SynN and DN in Figure 2.

## 5.2 Synergistic learning in adversarial environments

Statistics has a rich history of *robust learning* (Huber, 1996; Ramoni & Sebastiani, 2001), and machine learning has recently focused on *adversarial learning* (Szegedy et al., 2014; Zhang et al., 2018; 2020; Lowd & Meek, 2005). However, in both cases the focus is on adversarial *examples*, rather than adversarial *tasks*. In the context of synergistic learning, we informally define a task $t$ to be adversarial with respect to task $t'$ if the true joint distribution of task $t$, without any domain adaptation, impedes performance on task $t'$. In other words, training data from task $t$ can only add noise, rather than signal, for task $t'$. An adversarial task for Gaussian XOR is Gaussian XOR rotated by 45° (R-XOR) (Figure 2Aiii). Training on R-XOR therefore impedes the performance of SynF and SynN on XOR, and thus backward transfer becomes negative, demonstrating graceful forgetting (Aljundi et al., 2018) (Figure 2Ci). Because R-XOR is more difficult than XOR for SynF (because the discriminant boundaries are oblique (Tomita et al., 2020)), and because the discriminant boundaries are learned imperfectly with finite data, data from XOR can actually improve performance on R-XOR, and thus forward transfer is positive. In contrast, both forward and backward transfer are negative for RF and DN.

To further investigate this relationship, we design a suite of R-XOR examples, generalizing R-XOR from only 45° to any rotation angle between 0° and 90°, sampling 100 points from XOR, and another 100 from each R-XOR (Figure 2Cii). As the angle increases from 0° to 45°, log BLE flips from positive ($\approx 0.18$) to negative ($\approx -0.11$) for SynF. A similar trend is also visible for SynN. The 45°-XOR is the maximally adversarial R-XOR. Thus, as the angle further increases, log BLE increases back up to $\approx 0.18$ at 90°, which has an identical discriminant boundary to XOR. Moreover, when $\theta$ is fixed at 25°, BLE increases at different rates for different sample sizes of the source task (Figure 2Ciii).

Together, these experiments indicate that the amount of transfer can be a complicated function of (i) the difficulty of learning good representations for each task, (ii) the relationship between the two tasks, and (iii) the sample size of each. Appendix F further investigates this phenomenon in a multi-spiral environment.

## 6 Benchmark data experiments

For benchmark data, we build SynN encoders using the network architecture described in van de Ven et al. (2020) as "5 convolutional layers followed by two fully-connected layers each containing 2000 nodes with ReLU non-linearities and a softmax output layer". We use the same network architecture for all the benchmarking models as well. For the following experiments, we consider two modalities of real data: vision and language. Our language experiments in Appendix G.1 have qualitatively similar results as those of vision experiments illustrating that SynF and SynN are modality agnostic, sample and computationally efficient lifelong learning algorithms. In addition to the CIFAR 100 dataset, we provide additional vision experiments on larger datasets which show the relative performance gain of Model Zoo (boosting) (Ramesh & Chaudhari, 2021) compared to that of our approach (bagging) on large datasets. However, under the lifelong learning framework, a learning agent, constrained by capacity and computational time, is sequentially trained on multiple tasks. For each task, it has access to limited training samples (Chen & Liu, 2016; Lee et al., 2019; Kemker et al., 2018), and it improves on a particular task by leveraging knowledge from the other tasks. If a learner has enough single task data, it can achieve close to the optimal performance as a single task learner without any doing any sorts of transfer learning and thereby, will not be motivated to look for transfer of knowledge from other task data. Therefore, we are particularly interested in the behavior of our representation ensembling algorithms in the low training sample size regime using CIFAR 100 dataset. The CIFAR 10x10 experiments use only 500 training samples per task. For the corresponding experiments using higher training samples per task (5,000 samples), see Appendix Figure 4. For the FLE curves, we report forward learning efficiency on the corresponding task as that task is introduced. For backward learning efficiency, we evaluate the backward learning efficiency on all of the tasks introduced so far as a new task is introduced. Therefore, for each task the log(BLE) curve starts from 0 when the corresponding task is introduced and goes upward (positive) or downward (negative) as more tasks are seen. For getting an overall idea of the performance of the algorithms numerically, see Appendix Table 4, 5, 6 and 7 where we report the average log learning efficiency after all the tasks have been introduced for different algorithms as a summary. Apart

CIFAR 10X10 (500 samples)

Figure 3: **Performance of different algorithms on the CIFAR 10x10 vision experiments.** *Top left and middle*: Forward and backward learning efficiency for various resource building algorithms. SynF and SynN consistently demonstrate both forward and backward transfer for each task, whereas ProgNN and DF-CNN do not. In all of the plots, the performance of the chance algorithm which chooses a label at random is shown as a horizontal dashed line along 0. The positive and negative values are color coded with red and blue colors, respectively in the middle plot. *Bottom left and middle*: Same as above but comparing each algorithm with a fixed amount of resources. SynF is the only approach that demonstrate forward or backward transfer. *Top right*: Learning efficiencies of various algorithms for the 10 tasks after seeing the 10-th task. Both SynN and SynF synergistically learn over all the 10 tasks whereas other algorithms (except ProgNN) catastrophically forget. *Bottom right*: Building and recycling ensembles are two boundaries of a continuum, with hybrid models in the middle. SynF achieves lower (better) generalization error than other approaches until 5,000 training samples on the new task are available, but eventually a hybrid approach wins.

from our proposed metrics, we also report performance according to the metrics proposed by Veniat et al. (2020) in Figure 10 to verify the relevance of our proposed metrics.

## 6.1 CIFAR 10x10 dataset

The CIFAR 100 challenge (Krizhevsky, 2012), consists of 50,000 training and 10,000 test samples, each a 32x32 RGB image of a common object, from one of 100 possible classes, such as apples and bicycles. CIFAR 10x10 divides these data into 10 tasks, each with 10 classes (Lee et al., 2019) (see Appendix G for details). We compare SynF and SynN to the deep lifelong learning algorithms discussed above.

### 6.1.1 Resource growing experiments

We first compare SynF and SynN to several resource growing algorithms: Model Zoo, ProgNN(Rusu et al., 2016) and DF-CNN (Lee et al., 2019) (Figure 3, top panels). Both SynF and SynN demonstrate positive forward transfer for every task (SynF increases nearly monotonically), indicating they are robust to distributional shift in ways that ProgNN and DF-CNN are not. SynN, SynF and Model Zoo (Ramesh & Chaudhari, 2021) demonstrate positive backward transfer, SynN is actually monotonically increasing, indicating that with each new task, performance on all prior tasks increases (and SynF nearly monotonically increases BLE as well). In contrast, neither ProgNN nor DF-CNN exhibit any positive backward transfer. Final learning efficiency per task in the third row first plot is the learning efficiency associated with that task having seen all the data. SynF and SynN both demonstrate positive final learning efficiency for all tasks (synergistic learning), whereas ProgNN and DF-CNN both exhibit negative final learning efficiency for at least one task.

### 6.1.2 Resource constrained experiments

It is possible that the above algorithms are leveraging additional resources to improve performance without meaningfully transferring information between representations. To address this concern, we devised a "resource constrained" variant of SynF. In this constrained variant, we compare the lifelong learning algorithm to its single task variant, but ensure that they both have the same amount of resources. For example, on Task 2, we would compare SynF with 20 trees (10 trained on 500 samples from Task 1, and another 10 trained on 500 samples from Task 2) to RF with 20 trees (all trained on 500 samples Task 2). If SynF is able to meaningfully transfer information across tasks, then its resource-constrained FLE and BLE will still be positive. Indeed, FLE remains positive after enough tasks, and BLE is actually invariant to this change (Figure 3, bottom left and center). In contrast, all of the reference algorithms that have fixed resources exhibit negative forward and backward transfer. Moreover, the reference algorithms also all exhibit negative final transfer efficiency on each task, whereas our resource constrained SynF maintains positive final transfer on every task (Figure 3, top right). Interestingly, when using 5,000 samples per task, total and partial replay methods are able to demonstrate positive forward and backward transfer (Supplementary Figure 6), although they require quadratic time. Note that in this experiment, building the single task learners actually requires substantially *more* resources, specifically, $10 + 20 + \cdots + 100 = 550$ trees, as compared with only 100 trees in the prior experiments. In general, to ensure single task learners use the same amount of resources per task as omnidirectional learners requires $\tilde{\mathcal{O}}(n^2)$ resources, where as SynF only requires $\tilde{\mathcal{O}}(n)$, a polynomial reduction in resources.

In both cases, resource growing or resource constrained, both SynF and SynN show synergistic learning over all the 10 tasks (Figure 3, top right panel) whereas all other algorithms except Model Zoo and ProgNN suffer from catastrophic forgetting. The performance metrics in Figure 10 reveals the same story about the performance of the algorithms as that of our proposed metrics.

### 6.1.3 Resource Recycling Experiments

The binary distinction we made above, algorithms either build resources or reallocate them, is a false dichotomy, and biologically unnatural. In biological learning, systems develop from building (juvenile) to constrained (adult) resources (which requires recycling some resources for new tasks). We therefore train SynF on the first nine CIFAR 10x10 tasks using 50 trees per task, with 500 samples per task. For the tenth task, we could (i) select the 50 trees (out of the 450 existing trees) that perform best on task 10 (recruiting),

(ii) train 50 new trees, as `SynF` would normally do (building), (iii) build 25 and recruit 25 trees (hybrid), or (iv) ignore all prior trees (`RF`). `SynF` outperforms other approaches except when 5,000 training samples are available, but the recycling approach is nearly as good as `SynF` (Figure 3, bottom right). This result motivates future work to investigate optimal strategies for determining how to optimally leverage existing resources given a new task, and task-unaware settings.

### 6.1.4 Comparison with Single Task Experts

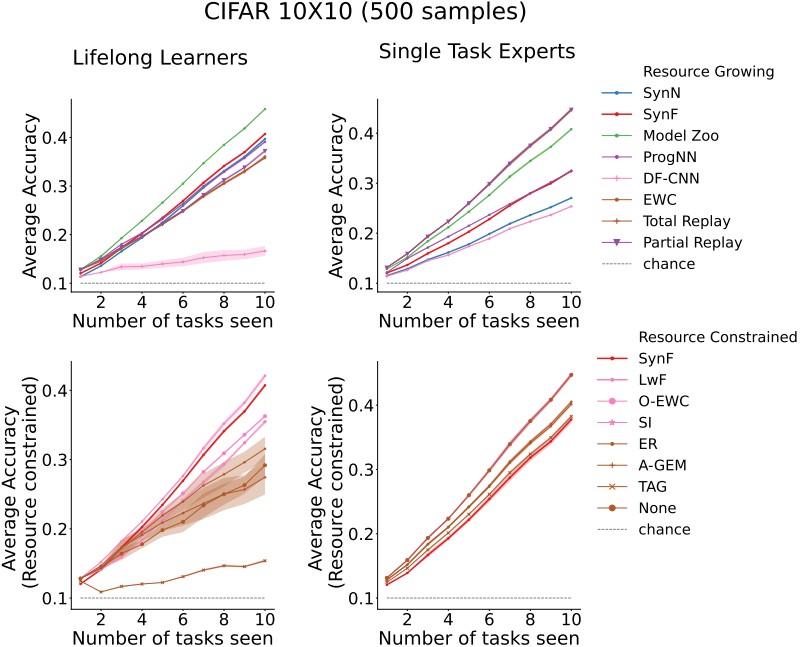

Figure 4: **Average accuracy over** 10 **tasks as the learners (lifelong and single task experts) see more tasks.** For single task learners, a new stand-alone learner is trained as a new task is seen and the average accuracy over all the task specific learners is reported. `LwF` has the highest multitask accuracy (bottom left) on CIFAR 10X10 while it the best single task accuracy and `SynF` has the lowest single task accuracy (bottom right). Therefore, only accuracy can falsely detect positive transfer. The error bar ($\pm 1.96 \times$ std) is shown as a faded color spread centering the mean curve.

In this experiment, we train the lifelong learning algorithms and the single task experts sequentially on 10 tasks from CIFAR 10X10. After each task is introduced, a new single task expert with the same hyperparameters as the corresponding lifelong learner is trained on the task and at the same time, the lifelong learning models are updated for the same task. The lifelong learners and their single task expert counterparts are shown using the same marker and color in Figure 4. Therefore, we will have 10 single task experts for each lifelong learners after 10 tasks are seen. As each new task is seen, we evaluate the performance of the algorithms on 10 tasks and report the average accuracy $< \mathcal{A}_t >$ (Lomonaco & Maltoni, 2017; Maltoni & Lomonaco, 2019) as:

$$< \mathcal{A}_t > = \frac{1}{T} \sum_{t'=1}^{T} (1 - \mathcal{E}_f^{t'}(\bigcup_{i=1}^{t} \mathbf{S}^i)) \tag{9}$$

Note that we have chance performance for the tasks not seen yet. If the lifelong learners are able to perform better than their single task expert counterpart, we say the learner successfully transferred knowledge between the tasks. As apparent from Figure 4, only multitask accuracy cannot ascertain the superiority of an algorithm. For example, note that in Figure 4 bottom left, `LwF` (Li & Hoiem, 2017) has better average accuracy compared to that of `SynF`. However, as shown in the bottom right of Figure 4, `LwF` has relatively higher single task accuracy compared to that of `SynF`. This is because `LwF` utilizes convolutional layers to

extract the local information in the image data while `SynF` cannot do that. Therefore, `LwF` improves accuracy for each task without doing meaningful transfer of information between the tasks. This is evident from the forward and the backward learning efficiency curves in the middle row of Figure 3.

### 6.1.5 Ablation Experiment

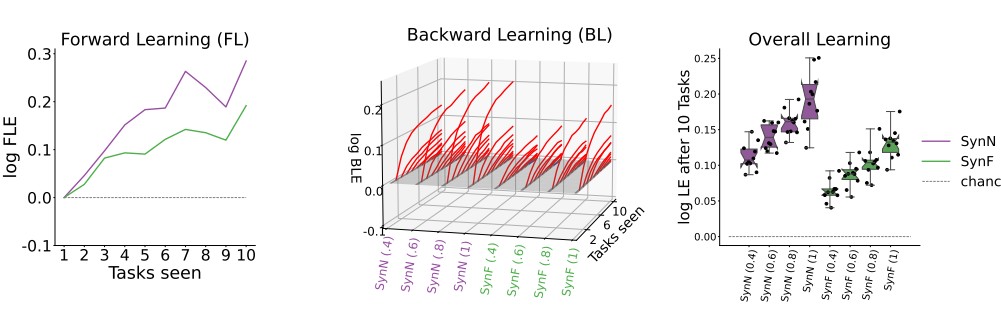

Figure 5: **Ablation experiment on CIFAR 10x10.** Fraction of total samples per task replayed is mentioned in parenthesis in the middle and the right plot. *Left:* `FLE` for each task remains the same for different amount of replay from the old tasks, i.e., the `FLE` curves for each algorithm with different amount of replay are superimposed on each other. *Middle:* Amount of backward transfer increases as more samples are replayed from the old tasks. The positive and negative values are color coded with red and blue colors, respectively. *Right:* With `FLE` remaining constant, the overall learning efficiency (`LE`) increases as the `BLEs` for different tasks increase with an increasing number of replayed samples.

In this experiment, we train 4 different versions of `SynN` and `SynF` sequentially on the 10 tasks from CIFAR 10X10. The only difference between different versions of the algorithms is the amount of old task data replayed. In 4 different versions of each algorithm, we replay 40%, 60%, 80% and 100% of the old task data respectively. As apparent from Figure 5, replaying old task data has no effect on forward transfer and the proposed algorithms transfer more as more data from old tasks are replayed.

### 6.1.6 Adversarial experiments

Consider the same CIFAR 10x10 experiments above, but, for tasks two through nine, randomly permute the class labels within each task, rendering each of those tasks adversarial with regard to the first task (because the labels are uninformative). Figure 6A indicates that `BLE` for both `SynF` and `SynN` is invariant to such label shuffling (the other algorithms also seem invariant to label shuffling, but did not demonstrate positive backward transfer). Now, consider a Rotated CIFAR experiment, which uses only data from the first task, divided into two equally sized subsets (making two tasks), where the second subset is rotated by different amounts (Figure 6, right). Learning efficiency of both `SynF` and `SynN` is nearly invariant to rotation angle, whereas the other approaches are far more sensitive to rotation angle. Note that zero rotation angle corresponds to the two tasks *having identical distributions*.

### 6.2 Five Dataset

In this experiment, we have used the **Five-dataset** provided in `https://github.com/pranshu28/TAG`. It consists of 5 tasks from five different dataset- CIFAR-10 (Krizhevsky, 2012), MNIST, SVHN (Netzer et al., 2011), notMNIST (Bulatov, 2011), Fashion-MNIST (Xiao et al., 2017). All the monochromatic images are converted to RGB format depending on the dataset. All images are then resized to $3 \times 32 \times 32$. As

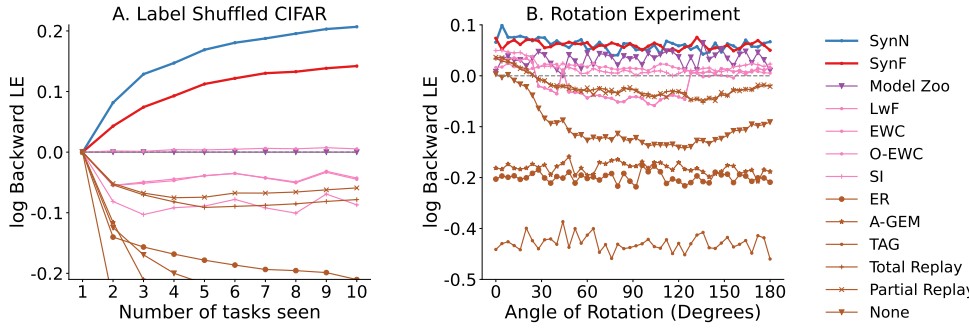

Figure 6: **Extended CIFAR 10x10 experiments.** *A*. Shuffling class labels within tasks two through nine with 500 samples each demonstrates both `SynF` and `SynN` can still achieve positive backward transfer, and that the other algorithms still fail to transfer. *B*. `SynF` and `SynN` are nearly invariant to rotations, whereas other approaches are more sensitive to rotation.

| Table 3: 5-dataset details. | | |
|---|---|---|
| | Training samples | Testing samples |
| CIFAR-10 | 50000 | 10000 |
| MNIST | 60000 | 10000 |
| SVHN | 73257 | 26032 |
| notMNSIT | 16853 | 1873 |
| Fashion-MNIST | 60000 | 10000 |

shown in table 3, training samples per task in 5-dataset is relatively higher than that of low data regime ideally considered in lifelong learning setting. However, `SynN` and `SynF` show less forgetting than most of the benchmarking algorithms. On the other hand, model zoo shows comparatively better performance in relatively high task data size setup.

### 6.3   Split Mini-Imagenet

In this experiment, we have used the **Mini-Imagenet** dataset provided in `https://www.kaggle.com/datasets/whitemoon/miniimagenet`. The dataset was split into 20 tasks each 5 each. Each task has 2400 training samples and 600 testing samples. In this case, we get positive FLE and BLE for both `SynN` and `SynF`. However, model zoo outperforms all the algorithms in this experiment.

### 6.4   FOOD1k Dataset

The datasets considered so far are of small scales. In this experiment, we use **Food1k** which is a large scale vision dataset consisting of 1000 food categories from Food2k Min et al. (2021). We split the 1000 classes into 50 tasks with 20 classes each. For each class, we randomly sampled 60 samples per class for training the models and used rest of the data for testing purpose. Note that so far `Model Zoo` performs the best among the reference resource growing models and `LwF` is the best performing resource constrained algorithm. Therefore, we choose `Model Zoo` and `LwF` as the reference models for the large scale experiment to avoid heavy computational cost. As shown in Figure 9, `SynN`  performs the best among all the models and the backward learning efficiency for the earlier tasks eventually saturates indicating the possibility of resource constrained stage as mentioned previously in the resource recycling experiments on CIFAR 10X10 vision dataset.

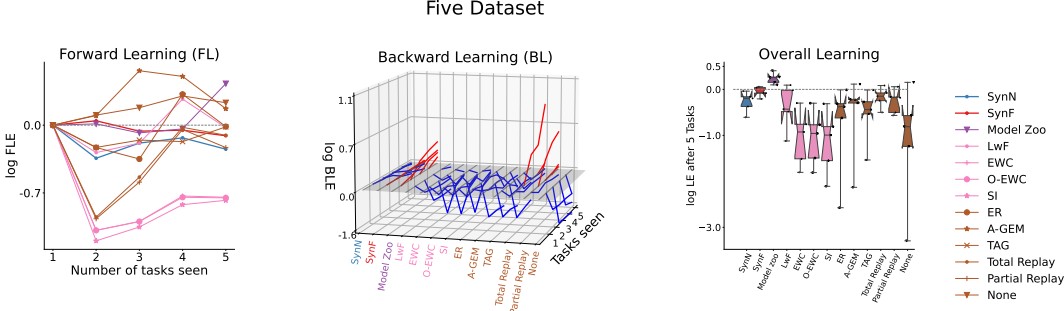

Figure 7: **Performance of different algorithms on the Five Datasets vision experiments.** The positive and negative values are color coded with red and blue colors, respectively in the top middle plot. Model Zoo performs the best and SynF performs the second best compared to all other algorithms in high sample size regimes (first row third panel). Sample size for each task is provided in Table 3.

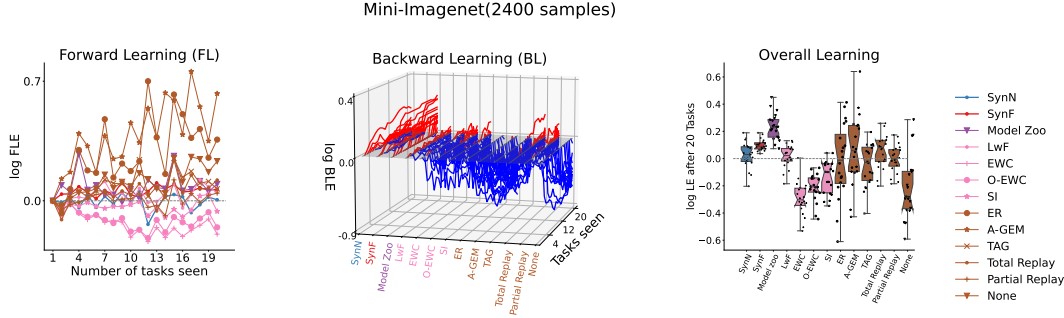

Figure 8: **Performance of different algorithms on the Split Mini-Imagenet vision experiments.** The positive and negative values are color coded with red and blue colors, respectively in the top middle plot. We have qulitatively similar results on Split Mini-Imagenet tasks as those of Five Dataset tasks. Note that each task in Mini-Imagnet has 2400 training samples which is lower than that of Five Dataset tasks. This relatively lower sample size results in a bit better performance for SynF and SynN compared to those on Five Dataset (first row third panel).

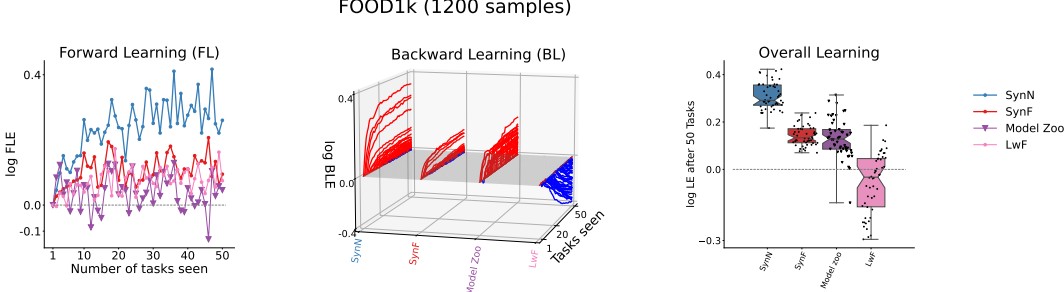

Figure 9: **Performance of different algorithms on the Food1k vision experiments.** The positive and negative values are color coded with red and blue colors, respectively in the top middle plot. SynN performs the best on large scale datasets like food1k.

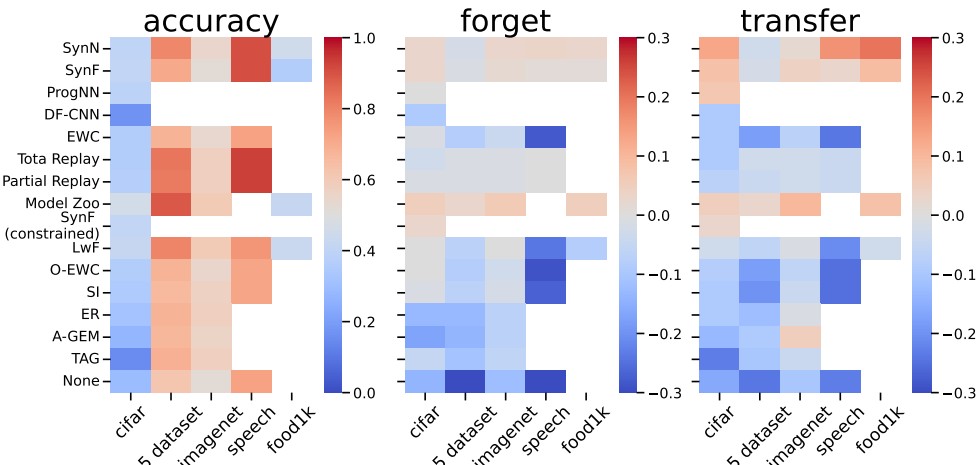

Figure 10: **Performance metrics: average accuracy, forgetting and average transfer as proposed by Veniat et al. (2020) calculated for different algorithms on different datasets.** SynN, SynF and Model Zoo show better accuracy, backward transfer (positive forgetting) and higher transfer among all other algorithms. Note that LwF has better accuracy on cifar dataset than that of our proposed methods, but it has negative transfer which was previously explained in Figure 4. The empty row for a benchmark data represents the algorithms that we could not run using the code provided by the authors.

## 7 Discussion

We introduced quasilinear representation ensembling as an approach to synergistic lifelong learning. Two specific algorithms, SynF and SynN, achieve both forward and backward transfer, due to leveraging resources (encoders) learned for other tasks without undue computational burdens. In this paper, we have mainly focused on task-aware setting as it is relatively easier to explain our approach and insights than other settings. In our future work, we will extend our approach to a more realistic task-unaware incremental class learning scenarios. Recycling experiment with CIFAR 10x10 shows that Forest-based representation ensembling approaches can easily add new resources when appropriate. This work therefore motivates additional work on deep learning to enable dynamically adding resources when appropriate and resuse the older representations like the moduler methods (Yoon et al., 2017; Mallya & Lazebnik, 2018; Veniat et al., 2020; Ostapenko et al., 2021).

To achieve backward transfer, SynF and SynN stored old data to vote on the newly learned transformers. Because the representation space scales quasilinearly with sample size, storing the data does not increase the space complexity of the algorithm, and it remains quasilinear. It could be argued that by keeping old data and training a model with increasing capacity from scratch (a sequential multitask learning approach), it would be straightforward to maintain performance (TE = 1) in a particular task. However, it is not obvious how to achieve backward transfer with quasilinear time and space complexity even if we are allowed to store all the past data, because computational time would naively become quadratic. For example, both ProgNN and Total Replay have quadratic time complexity, unlike SynF and SynN. Thus, one natural extension of this work that could mitigate the need to store all the data by using a generative model or subsampling.

While we employed quasilinear representation ensembling to address catastrophic forgetting, the paradigm of ensembling *representations* rather than *decisions* can be readily applied more generally. For example, "batch effects" (sources of variability unrelated to the scientific question of interest) have plagued many fields of inquiry, including neuroscience (Bridgeford et al., 2020) and genomics (Johnson et al., 2007). Similarly, federated learning is becoming increasingly central in artificial intelligence, due to its importance in differential privacy (Dwork, 2008). This may be particularly important in light of global pandemics such as COVID-19, where combining small datasets across hospital systems could enable more rapid discoveries (Vogelstein et al., 2020).

Finally, our quasilinear representation ensembling approach closely resembles the constructivist view of brain development (Quartz, 1999; Karmiloff-Smith, 2017). According to this view, the brain goes through progressive elaboration of neural circuits resulting in an augmented cognitive representation while maturing in a certain skill. In a similar way, representation ensembling algorithms can mature in a particular skill such as vision tasks by learning a rich encoder dictionary from different vision datasets and thereby, transfer forward to future or yet unseen vision dataset (see CIFAR 10x10 recruitment experiment as a proof). However, there is also substantial pruning during development and maturity in the brain circuitry which is important for performance (Sakai, 2020). This motivates future work for pruning adversarial encoders to enhance the transferability among tasks even more. Moreover, by carefully designing experiments in which both behaviors and brain are observed while learning across sequences of tasks (possibly in multiple stages of neural development or degeneration), we may be able to learn more about how biological agents are able to synergistically learn so efficiently, and transfer that understanding to building more effective artificial intelligences.

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

## A  Decision Tree as a Compositional Hypothesis

Consider learning a decision tree for a two class classification problem. The input to the decision tree is a set of $n$ feature-vector/response pairs, $(x_i, y_i)$. The learned tree structure corresponds to the encoder $u$, because the tree structure maps each input feature vector into an indicator encoding in which leaf node each feature vector resides. Formally, $u : \mathcal{X} \mapsto [L]$, where $[L] = \{\mathbb{1}_{\{\mathcal{X} \in l_1\}}, \mathbb{1}_{\{\mathcal{X} \in l_2\}}, \dots, \mathbb{1}_{\{\mathcal{X} \in l_L\}}\}$ and $L$ is the total number of leaf nodes. In other words, $u$ maps from the original data space, to a $L$-dimensional one-hot encoded sparse binary vector, where the sole non-zero entry indicates in which leaf node a particular observation falls, that is, $\tilde{x} := u(x) \in \{0, 1\}^L$ where $\|\tilde{x}\| = 1$.

Learning the channel is simply a matter of counting the fraction of observations in each leaf per class. So, the channel is trained using $n$ pairs of transformed feature-vector/response pairs $(\tilde{x}_i, y_i)$, and it assigns a probability of each class in each leaf: $v_l := \mathbb{P}[y_i = 1 | \tilde{x}_i = l], \forall l \in \{1, 2, \cdots, L\}$ and $v(\tilde{x}) = \bigcup_{l=1}^L v_l$. In other words, for two class classification, $v$ maps from the $L$-dimensional binary vector to the probability that $x$ is in class 1. The decider is simply $w(v(\tilde{x})) = \mathbb{1}_{\{v(\tilde{x}) > 0.5\}}$, that is, it outputs the most likely class label of the leaf node that $x$ falls into.

For inference, the tree is given a single $x$, and it is passed down the tree until it reaches a leaf node, where it is represented by its leaf identifier $\tilde{x}$. The channel takes $\tilde{x}$ as input, and outputs the estimated posterior probability of being in class 1 for the leaf node in which $\tilde{x}$ resides: $v(\tilde{x}) = \mathbb{P}[y = 1 | \tilde{x}]$. If $v(\tilde{x})$ is bigger than 0.5, the decider decides that $x$ is in class 1, and otherwise, it decides it is in class 0.

## B  Compositional Representation Ensembling

Consider a scenario in which we have two tasks, one following the other. Assume that we already learned a single decomposable hypothesis for the first task: $w_1 \circ v_1 \circ u_1$, and then we get new data associated with a second task. Let $n_1$ denote the sample size for the first task, and $n_2$ denote the sample size for the second task, and $n = n_1 + n_2$. The representation ensembling approach generally works as follows. First, since we want to transfer forward to the second task, we push all the new data through the first encoder $u_1$, which yields $\tilde{x}_{n_1+1}^{(1)}, \dots, \tilde{x}_n^{(1)}$. Second, we learn a new encoder $u_2$ using the new data, $\{(x_i, y_i)\}_{i=n_1+1}^n$. We then push the new data through the new encoder, yielding $\tilde{x}_{n_1+1}^{(2)}, \dots, \tilde{x}_n^{(2)}$. Third, we train a new channel, $v_2$. To do so, $v_2$ is trained on the outputs from both encoders, that is, $\{(\tilde{x}_i^{(j)}, y_i)\}_{i=n_1+1}^n$ for $j = 1, 2$. The output of $v_2$ for any new input $x$ is the posterior probability (or score) for that point for each potential response in task two (class label). Thus, by virtue of ensembling these representations, this approach enables forward transfer (Rusu et al., 2016; Dhillon et al., 2020).

Now, we would also like to improve performance on the first task using the second task's data. While many lifelong methods have tried to achieve this kind of backward transfer, to date, they have mostly failed (Ruvolo & Eaton, 2013). Recall that previously we had already pushed all the first task data through the first task encoder, which had yielded $\tilde{x}_1^{(1)}, \dots, \tilde{x}_{n_1}^{(1)}$. Assuming we kept any of the first task's data, or can adequately simulate it, we can push those data through $u_2$ to get a second representation of the first task's data: $\tilde{x}_1^{(2)}, \dots, \tilde{x}_{n_1}^{(2)}$. Then, $v_1$ would be trained on both representations of the first task's data. This 'replay-like' procedure facilitates backward transfer, that is, improving performance on previous tasks by leveraging data from newer tasks. Both the forward and backward transfer updates can be implemented every time we obtain data associated with a new task. Enabling the channels to ensemble *omnidirectionally* between all sets of tasks is the key innovation of our proposed synergistic learning approaches.

## C  Synergistic Algorithms

In this paper, we have proposed two concrete synergistic algorithms, Synergistic Forests (SynF) and Synergistic Networks (SynN). The two algorithms differ in their details of how to update encoders and channels, but abstracting a level up they are both special cases of the same procedure. Let SynX refer to any possible synergistic algorithm. Algorithms 1, 2, 3, and 4 provide pseudocode for adding encoders, updating channels, and making predictions for any SynX algorithm. Whenever the learner gets access to a new task data, we

---

**Algorithm 1** Add a new SYNX encoder for a task. OOB = out-of-bag.

---

**Require:**
 (1) $t$                     ▷ current task number
 (2) $\mathcal{D}_n^t = (\mathbf{x}^t, \mathbf{y}^t) \in \mathbb{R}^{n \times p} \times \{1, \ldots, K\}^n$         ▷ training data for task $t$
**Ensure:**
 (1) $u_t$                    ▷ an encoder trained on task $t$
 (2) $\mathcal{I}_{OOB}^t$                 ▷ a set of the indices of OOB data
 1: **function** SYNX.FIT$(t, (\mathbf{x}^t, \mathbf{y}^t))$
 2:    $u_t, \mathcal{I}_{OOB}^t \leftarrow$ encoder.fit$(\mathbf{x}^t, \mathbf{y}^t)$       ▷ train an encoder on partitioned data
 3:    **return** $u_t, \mathcal{I}_{OOB}^t$
 4: **end function**

---

**Algorithm 2** Add a new SYNX channel for the current task.

---

**Require:**
 (1) $t$                     ▷ current task number
 (2) $\boldsymbol{u}_t = \{u_t\}_{t'=1}^t$               ▷ the set of encoders
 (3) $\mathcal{D}_n^t = (\mathbf{x}^t, \mathbf{y}^t) \in \mathbb{R}^{n \times p} \times \{1, \ldots, K\}^n$        ▷ training data for task $t$
 (4) $\mathcal{I}_{OOB}^t$         ▷ a set of the indices of OOB data for the current task
**Ensure:** $\boldsymbol{v}_t = \{v_{t,t'}\}_{t'=1}^t$      ▷ in-task $(t' = t)$ and cross-task $(t' \neq t)$ channels for task $t$
 1: **function** SYNX.ADD_CHANNEL$(t, \boldsymbol{u}_t, (\mathbf{x}_t, \mathbf{y}_t), \mathcal{I}_{OOB}^t)$
 2:    $v_{tt} \leftarrow u_t$.add_channel$((\mathbf{x}_t, \mathbf{y}_t), \mathcal{I}_{OOB}^t)$     ▷ add the in-task channel using OOB data
 3:    **for** $t' = 1, \ldots, t-1$ **do**       ▷ update the cross task channels for task $t$
 4:      $v_{tt'} \leftarrow u_{t'}$.add_channel$(\mathbf{x}_t, \mathbf{y}_t)$
 5:    **end for**
 6:    **return** $v_t$
 7: **end function**

---

use Algorithm 1 to train a new encoder for the corresponding task. We split the data into two portion–
one set is used to learn the encoder and the other portion is called the held out or out-of-bag (OOB) data
which is returned by Algorithm 1 to be used by Algorithm 3 to learn the channel for the corresponding
task. Note that we push the OOB data through the in-task encoder and the whole dataset through the
cross-task encoders to update the channel, i.e, learn the posteriors according to the new encoder. Then
we use Algorithm 3 to replay the old task data through the new encoder and update their corresponding
channels. Finally, while predicting for a test sample, we use Algorithm 4. Given the task identity, we use
the corresponding channel to get the average estimated posterior and predict the class label as the arg max
of the estimated posteriors.

Table 1: Hyperparameters for SYNF in CIFAR-10X10 experiments. n_estimators is denoted by $B$, the
number of trees, above.

| Hyperparameters | Value |
|---|---|
| n_estimators (500 training samples per task) | 10 |
| n_estimators (5000 training samples per task) | 40 |
| max_depth | 30 |
| max_samples (OOB split) | 0.67 |
| min_samples_leaf | 1 |

## D    Reference Algorithm Implementation Details

The same network architecture was used for all compared deep learning methods. Following van de Ven et al.
(2020), the 'base network architecture' consisted of five convolutional layers followed by two-fully connected

---

**Algorithm 3** Update SYNX channel for the previous tasks.

---
**Require:**
    (1) $t$          ▷ current task number
    (2) $u_t$          ▷ encoder for the current task
    (3) $\mathcal{D} = \{\mathcal{D}^{t'}\}_{t'=1}^{t-1}$          ▷ training data for tasks $t' = 1, \cdots, t-1$
**Ensure:** $\boldsymbol{v} = \{\boldsymbol{v}_{t'}\}_{t'=1}^{t-1}$          ▷ all previous task voters
  1: **function** SYNX.UPDATE_CHANNEL($t, u_t, \mathcal{D}$)
  2:     **for** $t' = 1, \ldots, t-1$ **do**          ▷ update the cross task channels
  3:         $v_{t't} \leftarrow u_t.\text{get\_channel}(\mathbf{x}_{t'}, \mathbf{y}_{t'})$
  4:     **end for**
  5:     **return** $\boldsymbol{v}$
  6: **end function**

---

**Algorithm 4** Predicting a class label using SYNX.

---
**Require:**
    (1) $x \in \mathbb{R}^p$          ▷ test datum
    (2) $t$          ▷ task identity associated with $x$
    (3) $\boldsymbol{u}$          ▷ all $T$ reperesenters
    (4) $\boldsymbol{v}_t$          ▷ channel for task $t$
**Ensure:** $\hat{y}$          ▷ a predicted class label
  1: **function** $\hat{y} = $ SYNX.PREDICT($t, x, v_t$)
  2:     $T \leftarrow$ SYNX.get\_task\_number()          ▷ get the total number of tasks
  3:     $\hat{\mathbf{p}}_t = \mathbf{0}$          ▷ $\hat{\mathbf{p}}_t$ is a $K$-dimensional posterior vector
  4:     **for** $t' = 1, \ldots, T$ **do**          ▷ aggregate the posteriors calculated from $T$-th task channel
  5:         $\hat{\mathbf{p}}_t \leftarrow \hat{\mathbf{p}}_t + v_{tt'}.\text{predict\_proba}(u_{t'}(x))$
  6:     **end for**
  7:     $\hat{\mathbf{p}}_t \leftarrow \hat{\mathbf{p}}_t / T$
  8:     $\hat{y} = \arg\max_i(\hat{\mathbf{p}}_t)$          ▷ find the index $i$ of the elements in the vector $\hat{\mathbf{p}}_t$ with maximum probability
  9:     **return** $\hat{y}$
10: **end function**

---

layers each containing 2000 nodes with ReLU non-linearities and a softmax output layer. The convolutional layers had 16, 32, 64, 128 and 254 channels, they used batch-norm and a ReLU non-linearity, they had a 3x3 kernel, a padding of 1 and a stride of 2 (except the first layer, which had a stride of 1). This architecture was used with a multi-headed output layer (i.e., a different output layer for each task) for all algorithms using a fixed-size network. For ProgNN and DF-CNN the same architecture was used for each column introduced for each new task, and in our SYNN this architecture was used for the transformers $u_t$ (see above). In these implementations, ProgNN and DF-CNN have the same architecture for each column introduced for each task. Among the reference algorithms, EWC, O-EWC, LwF, SI, TOTAL REPLAY and PARTIAL REPLAY results were produced using the repository `https://github.com/GMvandeVen/progressive-learning-pytorch`. For ProgNN and DF-CNN we used the code provided in `https://github.com/Lifelong-ML/DF-CNN`. For all other reference algorithms, we modified the code provided by the authors to match the deep net architecture as mentioned above and used the default hyperparameters provided in the code.

## E   Training Time Complexity Analysis

Consider a lifelong learning environment with $T$ tasks each with $n'$ samples, i.e., total training samples, $n = n'T$. For all the algorithm with time complexity $\tilde{\mathcal{O}}(n)$, the training time grows linearly with more training samples. We discuss all other algorithms with non-linear time complexity below.

Table 2: Hyperparameters for SYNF in Five Datasets, Split Mini-Imagenet, FOOD1k experiments. n_estimators is denoted by $B$, the number of trees, above. Note that we use the same hyperparameters for all of the aforementioned datasets.

| Hyperparameters | Value |
|---|---|
| n_estimators | 10 |
| max_depth | 30 |
| max_samples (OOB split) | 0.67 |
| min_samples_leaf | 1 |

Table 3: Hyperparameters for SYNN in CIFAR 10X10, Five Datasets, Split Mini-Imagenet, FOOD1k experiments. Note that we use the same hyperparameters for all of the aforementioned datasets.

| Hyperparameters | Value |
|---|---|
| optimizer | Adam |
| learning rate | $3 \times 10^{-4}$ |
| max_samples (OOB split) | 0.67 |
| K (KNN channel) | $\log_2(\text{number of samples per task})$ |

### E.1   EWC

Consider the time required to train the weights for each task in EWC is $k_c n'$ and each task adds additional $k_l n'$ time from the regularization term. Here, $k_c$ and $k_l$ are both constants. Therefore, time required to learn all the $T$ tasks can be written as:

$$
\begin{aligned}
k_c n' &+ (k_c n' + k_l n') + \cdots + (k_c n' + (T-1)k_l n') \\
&= k_c n' T + k_l n' \sum_{t=1}^{T-1} t \\
&= k_c n' T + k_l n' \frac{T(T-1)}{2} \\
&= k_c n + 0.5 k_l nT - 0.5 k_l n \\
&= \tilde{\mathcal{O}}(nT).
\end{aligned}
\tag{10}
$$

### E.2   Total Replay

Consider the time to train the model on $n'$ samples is $k_c n'$. Therefore, time required to learn all the $T$ tasks can be written as:

$$
\begin{aligned}
k_c n' &+ k_c(n' + n') + \cdots + k_c n' T \\
&= k_c n' \sum_{t=1}^{T} t \\
&= k_c n' \frac{T(T+1)}{2} \\
&= 0.5 k_c nT + 0.5 k_c n \\
&= \tilde{\mathcal{O}}(nT)
\end{aligned}
\tag{11}
$$

### E.3  ProgNN

Consider the time required to train each column in ProgNN is $k_c n'$ and each lateral connection can be learned with time $k_l n'$. Therefore, time required to learn all the $T$ tasks can be written as:

$$
\begin{aligned}
k_c n' &+ (k_c n' + k_l n') + \cdots + (k_c n' + (T-1)k_l n') \\
&= k_c n' T + k_l n' \sum_{t=1}^{T-1} t \\
&= k_c n' T + k_l n' \frac{T(T-1)}{2} \\
&= k_c n + 0.5 k_l n T - 0.5 k_l n \\
&= \tilde{\mathcal{O}}(nT)
\end{aligned}
\tag{12}
$$

## F  Simulated Results

In each simulation, we constructed an environment with two tasks. For each, we sample 750 times from the first task, followed by 750 times from the second task. These 1,500 samples comprise the training data. We sample another 1,000 hold out samples to evaluate the algorithms. We fit a random forest (`RF`) (technically, an uncertainty forest which is an honest forest with a finite-sample correction (Mehta et al., 2019)) and a `SynF`. We repeat this process 30 times to obtain errorbars. Error bars in all cases were negligible.

### F.1  Gaussian XOR

Gaussian XOR is two class classification problem with equal class priors. Conditioned on being in class 0, a sample is drawn from a mixture of two Gaussians with means $\pm \begin{bmatrix} 0.5, & 0.5 \end{bmatrix}^\mathsf{T}$, and variances proportional to the identity matrix. Conditioned on being in class 1, a sample is drawn from a mixture of two Gaussians with means $\pm \begin{bmatrix} 0.5, & -0.5 \end{bmatrix}^\mathsf{T}$, and variances proportional to the identity matrix. Gaussian XNOR is the same distribution as Gaussian XOR with the class labels flipped. Rotated XOR (R-XOR) rotates XOR by $\theta°$ degrees.

### F.2  Spirals

A description of the distributions for the two tasks is as follows: let $K$ be the number of classes and $S \sim$ multinomial$(\frac{1}{K}\vec{1}_K, n)$. Conditioned on $S$, each feature vector is parameterized by two variables, the radius $r$ and an angle $\theta$. For each sample, $r$ is sampled uniformly in $[0,1]$. Conditioned on a particular class, the angles are evenly spaced between $\frac{4\pi(k-1)t_K}{K}$ and $\frac{4\pi(k)t_K}{K}$ where $t_K$ controls the number of turns in the spiral. To inject noise along the spiral, we add Gaussian noise to the evenly spaced angles $\theta' : \theta = \theta' + \mathcal{N}(0, \sigma_K^2)$. The observed feature vector is then $(r \cos(\theta), r \sin(\theta))$. In Figure 1 we set $t_3 = 2.5$, $t_5 = 3.5$, $\sigma_3^2 = 3$ and $\sigma_5^2 = 1.876$.

Consider an environment with a three spiral and five spiral task (Figure 1). In this environment, axis-aligned splits are inefficient, because the optimal partitions are better approximated by irregular polytopes than by the orthotopes provided by axis-aligned splits. The three spiral data helps the five spiral performance because the optimal partitioning for these two tasks is relatively similar to one another, as indicated by positive forward transfer. This is despite the fact that the five spiral task requires more fine partitioning than the three spiral task. Because `SynF` grows relatively deep trees, it over-partitions space, thereby rendering tasks with more coarse optimal decision boundaries useful for tasks with more fine optimal decision boundaries. The five spiral data also improves the three spiral performance.

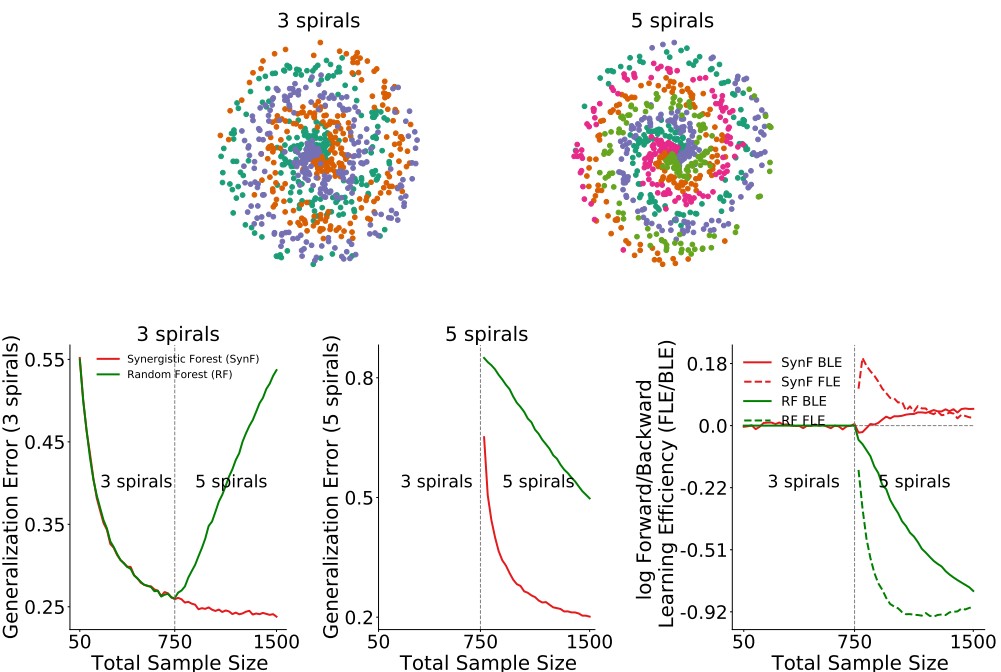

Figure 1: *Top*: 750 samples from 3 spirals (left) and 5 spirals (right). *Bottom left*: SynF outperforms RF on 3 spirals when 5 spirals data is available, demonstrating *backward* transfer in SynF. *Bottom center*: SynF outperforms RF on 5 spirals when 3 spirals data is available, demonstrating *forward* transfer in SynF. *Bottom right*: Transfer Efficiency of SynF. The forward (solid) and backward (dashed) curves are the ratio of the generalization error of SynF to RF in their respective figures. SynF demonstrates decreasing forward transfer and increasing backward transfer in this environment.

Table 4: **Performance metric: average** $\log(\mathsf{LE})$ **after** 10 **tasks calculated for different algorithms on CIFAR 10x10 (**500 **samples per task).**

| Algorithms | $< \log(\mathsf{LE}) > (\pm\text{std})$ |
|---|---|
| SynN | **0.19**($\pm$0.04) |
| SynF | **0.13**($\pm$0.02) |
| Model Zoo | 0.09($\pm$0.04) |
| ProgNN | 0.11($\pm$0.09) |
| DF-CNN | $-0.11$($\pm$0.08) |
| EWC | $-0.15$($\pm$0.04) |
| Total Replay | $-0.15$($\pm$0.03) |
| Partial Replay | $-0.13$($\pm$0.03) |
| SynF(resource constrained) | **0.05**($\pm$0.03) |
| LwF | $-0.05$($\pm$0.03) |
| O-EWC | $-0.14$($\pm$0.04) |
| SI | $-0.16$($\pm$0.03) |
| ER | $-0.13$($\pm$0.12) |
| A-GEM | $-0.19$($\pm$0.14) |
| TAG | $-0.32$($\pm$0.04) |
| None | $-0.24$($\pm$0.10) |

Table 5: **Performance metric: average** $\log(\text{LE})$ **after** 10 **tasks calculated for different algorithms on Five Dataset.**

| Algorithms | $< \log(\text{LE}) > (\pm\text{std})$ |
|---|---|
| SynN | $-\mathbf{0.27}(\pm 0.22)$ |
| SynF | $-\mathbf{0.05}(\pm 0.11)$ |
| Model Zoo | $0.24(\pm 0.12)$ |
| EWC | $-1.06(\pm 0.60)$ |
| Total Replay | $-0.18(\pm 0.22)$ |
| Partial Replay | $-0.27(\pm 0.26)$ |
| LwF | $-0.39(\pm 0.48)$ |
| O-EWC | $-1.07(\pm 0.60)$ |
| SI | $-1.15(\pm 0.69)$ |
| ER | $-0.78(\pm 1.03)$ |
| A-GEM | $-0.55(\pm 0.90)$ |
| TAG | $-0.56(\pm 0.58)$ |
| None | $-1.15(\pm 1.30)$ |

Table 6: **Performance metric: average** $\log(\text{LE})$ **after** 10 **tasks calculated for different algorithms on Mini-Imagenet.**

| Algorithms | $< \log(\text{LE}) > (\pm\text{std})$ |
|---|---|
| SynN | $\mathbf{0.02}(\pm 0.10)$ |
| SynF | $\mathbf{0.10}(\pm 0.04)$ |
| Model Zoo | $0.23(\pm 0.10)$ |
| EWC | $-0.29(\pm 0.12)$ |
| Total Replay | $0.06(\pm 0.13)$ |
| Partial Replay | $0.00(\pm 0.10)$ |
| LwF | $0.02(\pm 0.08)$ |
| O-EWC | $-0.21(\pm 0.10)$ |
| SI | $-0.14(\pm 0.12)$ |
| ER | $-0.02(\pm 0.27)$ |
| A-GEM | $0.06(\pm 0.26)$ |
| TAG | $-0.05(\pm 0.15)$ |
| None | $-0.22(\pm 0.23)$ |

## G   Real Data Extended Results

### G.1   Spoken Digit experiment

In this experiment, we used the **Spoken Digit** dataset provided in `https://github.com/Jakobovski/free-spoken-digit-dataset`. The dataset contains audio recordings from 6 different speakers with 50 recordings for each digit per speaker (3000 recordings in total). The experiment was set up with 6 tasks where each task contains recordings from only one speaker. For each recording, a spectrogram was extracted using Hanning windows of duration 16 ms with an overlap of 4 ms between the adjacent windows. The spectrograms were resized down to $28 \times 28$. The extracted spectrograms from 8 random recordings of '5' for 6 speakers are shown in Figure 2. For each Monte Carlo repetition of the experiment, spectrograms extracted for each task were randomly divided into 55% train and 45% test set. We have provided benchmarking for seven algorithms out of the 11 algorithms as mentioned in Subsection 2.3. As shown in Figure 3 and main text Figure 10, both SynF and SynN show positive transfer and synergistic learning between the spoken digit tasks, in contrast to other methods, some of which show only forward transfer, others show only backward transfer, with none showing both, and some showing neither.

Short-Time Fourier Transform Spectrogram of Number 5

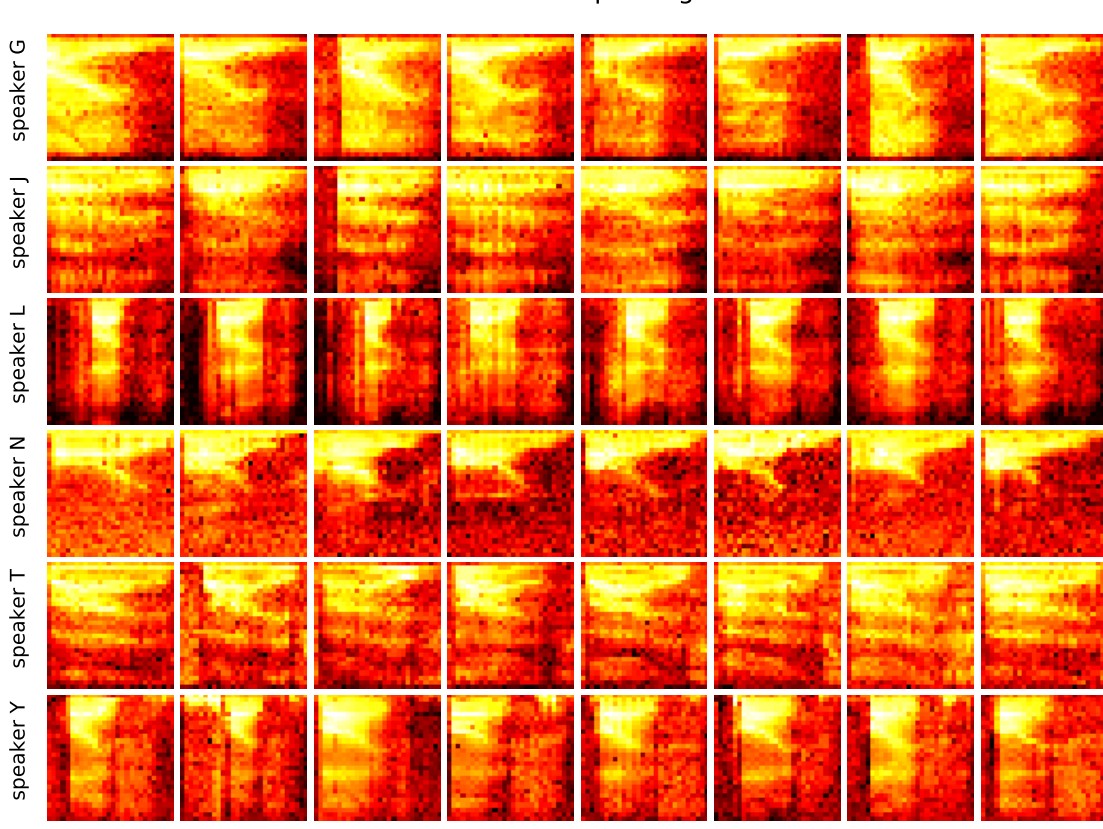

Figure 2: Spectrogram extracted from 8 different recordings of 6 speakers uttering the digit '5'.

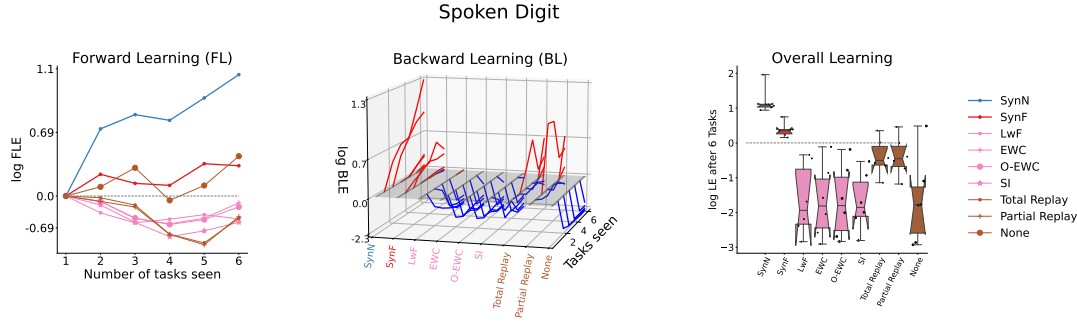

Figure 3: **Performance of different algorithms on the Spoken Digit experiments.** The positive and negative values are color coded with red and blue colors, respectively in the top middle panels. Both SynF and SynN show positive forward and backward transfer as well as synergistic learning for the spoken digit tasks, in contrast to other seven methods, some of which show only forward transfer, others show only backward transfer, with none showing both, and some showing neither.

Table 7: **Performance metric: average** $\log(\text{LE})$ **after** 10 **tasks calculated for different algorithms on FOOD1k.**

| Algorithms | $< \log(\text{LE}) > (\pm \text{std})$ |
|---|---|
| SYNN | **0.31**($\pm 0.06$) |
| SYNF | **0.14**($\pm 0.04$) |
| MODEL ZOO | 0.13($\pm 0.08$) |
| LwF | $-0.06$($\pm 0.13$) |

Table 8: Hyperparameters for SYNF in spoken digit experiment.

| Hyperparameters | Value |
|---|---|
| n_estimators (275 training samples per task) | 10 |
| max_depth | 30 |
| max_samples (OOB split) | 0.67 |
| min_samples_leaf | 1 |

## G.2 CIFAR 10x10

Supplementary Table 9 shows the image classes associated with each task number. Supplementary Figure 4 is the same as Figure 3 but with 5,000 training samples per task, rather than 500. Notably, with 5,000 samples, replay methods and Model Zoo are able to transfer both forward and backward as well. However, note that although total replay outperforms both SYNF and SYNN with large sample sizes, it is not a *bona fide* lifelong learning algorithm, because it requires $n^2$ time. Moreover, the replay methods will eventually forget as more tasks are introduced because it will run out of capacity.

## G.3 CIFAR Label Shuffling

Supplementary Figure 5 shows the same result as the label shuffling from Figure 6, but with 5,000 samples per class. The results for SYNN and SYNF are qualitatively similar, in that they transfer backward. The replay methods are also able to transfer when using this larger number of samples, although with considerably higher computational cost.

## G.4 CIFAR 10x10 Repeated Classes

We also considered the setting where each task is defined by a random sampling of 10 out of 100 classes with replacement. This environment is designed to demonstrate the effect of tasks with shared subtasks, which is

Table 9: Task splits for CIFAR 10x10.

| Task # | Image Classes |
|---|---|
| 1 | apple, aquarium fish, baby, bear, beaver, bed, bee, beetle, bicycle, bottle |
| 2 | bowl, boy, bridge, bus, butterfly, camel, can, castle, caterpillar |
| 3 | chair, chimpanzee, clock, cloud, cockroach, couch, crab, crocodile, cup, dinosaur |
| 4 | dolphin, elephant, flatfish, forest, fox, girl, hamster, house, kangaroo, keyboard |
| 5 | lamp, lawn mower, leopard, lion, lizard, lobster, man, maple tree, motor cycle, mountain |
| 6 | mouse, mushroom, oak tree, orange, orchid, otter, palm tree, pear, pickup truck, pine tree |
| 7 | plain, plate, poppy, porcupine, possum, rabbit, raccoon, ray, road, rocket |
| 8 | rose, sea, seal, shark, shrew, skunk, skyscraper, snail, snke, spider |
| 9 | squirrel, streetcar, sunflower, sweet pepper, table, tank, telephone, television, tiger, tractor |
| 10 | train, trout, tulip, turtle, wardrobe, whale, willow tree, wolf, woman, worm |

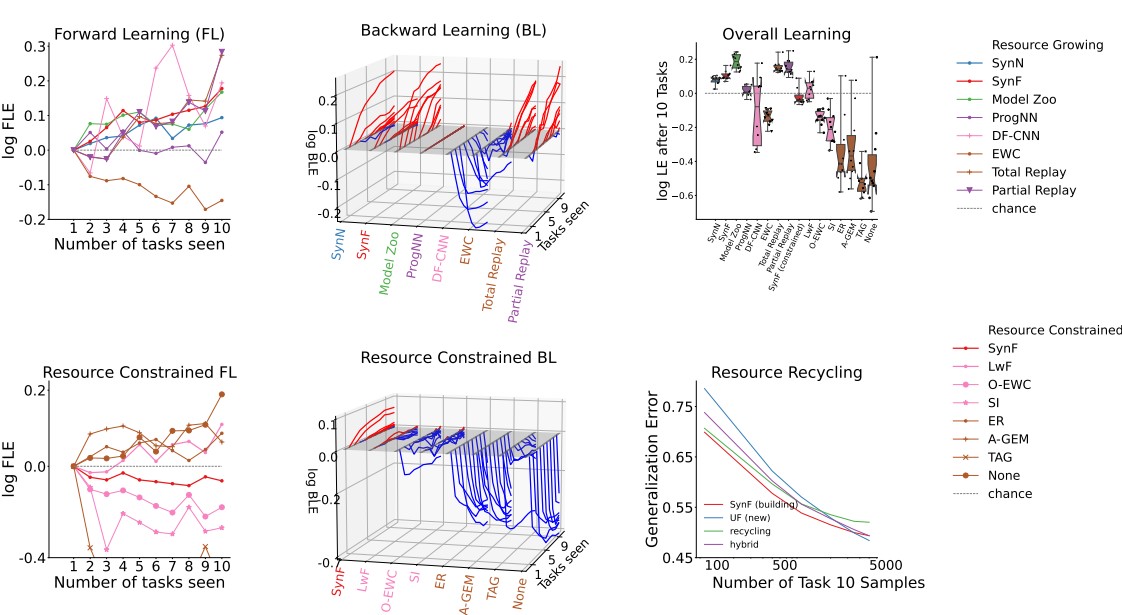

Figure 4: Performance of different algorithms on CIFAR 10x10 vision dataset for 5,000 training samples per task. SynN maintains approximately the same forward transfer (top left and middle left) and backward transfer (top and middle row second column) efficiency as those for 500 samples per task whereas other algorithms show reduced or nearly unchanged transfer. SynF still demonstrates positive forward, backward, and final transfer, unlike most of the state-of-the-art algorithms, which demonstrate forgetting. The replay methods, however, do demonstrate transfer, albeit with significantly higher computational cost.

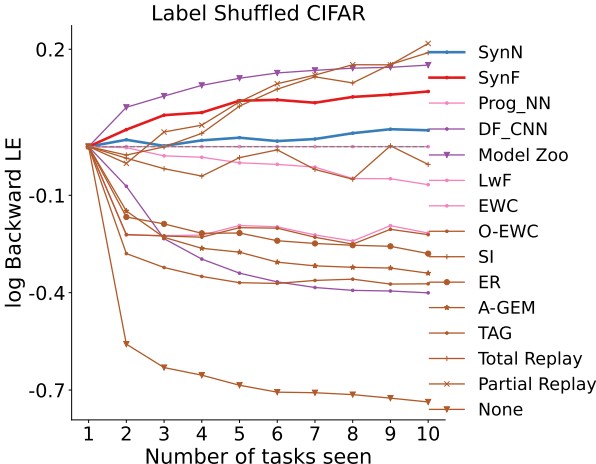

Figure 5: Label shuffle experiment on CIFAR 10x10 vision dataset for 5,000 training samples per task. Shuffling class labels within tasks two through nine with 5000 samples each demonstrates both SYNF and SYNN can still achieve positive backward transfer, and that the other algorithms that do not replay the previous task data fail to transfer.

Table 10: **Performance metrics: average accuracy $< \mathcal{A} >$, forgetting $< \mathcal{F} >$ and average transfer $< \mathcal{T} >$ as proposed by Veniat et al. (2020) calculated for different algorithms on CIFAR 10x10 ($5000$ samples per task).**

| Algorithms | $< \mathcal{A} >$ | $< \mathcal{F} >$ | $< \mathcal{T} >$ |
|---|---|---|---|
| SYNN | **0.95** | **0.0** | **0.0** |
| SYNF | **0.95** | **0.0** | **0.0** |
| ProgNN | 0.97 | 0.0 | 0.0 |
| DF-CNN | 0.93 | $-0.02$ | $-0.01$ |
| EWC | 0.95 | 0.0 | $-0.01$ |
| TOTAL REPLAY | 0.96 | 0.0 | 0.0 |
| PARTIAL REPLAY | 0.96 | 0.0 | 0.0 |
| MODEL ZOO | 0.96 | 0.01 | 0.01 |
| SYNF(resource constrained) | **0.95** | **0.0** | **0.0** |
| LwF | 0.96 | 0.0 | 0.0 |
| O-EWC | 0.95 | 0.0 | $-0.01$ |
| SI | 0.95 | 0.0 | $-0.01$ |
| ER | 0.94 | $-0.02$ | $-0.01$ |
| A-GEM | 0.94 | $-0.02$ | $-0.01$ |
| TAG | 0.92 | $-0.01$ | $-0.03$ |
| NONE | 0.93 | $-0.03$ | $-0.02$ |

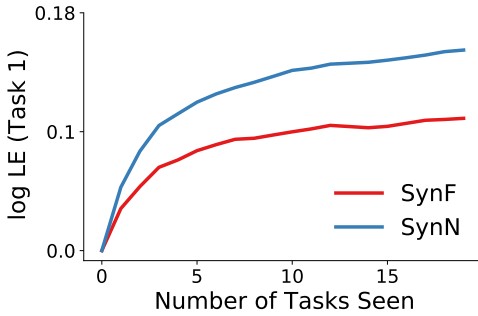

Figure 6: SYNF and SYNN transfer knowledge effectively when tasks share common classes. Each task is a random selection of 10 out of the 100 CIFAR-100 classes. Both SYNF and SYNN demonstrate monotonically increasing transfer efficiency for up to 20 tasks.

a common property of real world lifelong learning tasks. Supplementary Figure 6 shows transfer efficiency of SYNF and SYNN on Task 1.

