# OpenReview forum: "Representation Ensembling for Synergistic Lifelong Learning with Quasilinear Complexity"
_TMLR — Rejected by TMLR_

### Review · Reviewer_szZq · 2023-01-09

**Summary Of Contributions:**

In this work, the authors propose a method for lifelong learning based on ensembling representations from different tasks. They start by introducing new metrics for assessing performance of continual learning agents. Next, they introduce two algorithms based on the principle of representation ensembling - SynN which is based on deep neural networks and SynF which is based on decision forests. Next, the authors evaluate these approaches on a wide variety of benchmarks and compare them against existing continual learning methods. They show that in some settings the proposed approaches outperform other methods.

**Audience:**

Yes

**Broader Impact Concerns:**

I do not think a Broader Impact Statement is needed here.

**Claims And Evidence:**

No

**Requested Changes:**

 Overall I think the contents of the paper are interesting. However, certain parts of empirical evaluation are lacking and should be improved and I would still like to see several points addressed before explicitly recommending the paper for acceptance. I'd be satisfied with discussing them explicitly as limitations or topics for future work, especially in cases where fully addressing the problem would be costly in terms of time and computation.

In particular:
- In terms of evaluation, I think including more traditional metrics existing in related work would be useful for the reader. Additionally, I think the potential downsides of introduced metrics should be discussed.
- Ideally, a more recent baseline than progressive networks should be used. If this is not possible, a more thorough discussion of literature and a clear statement of limitations is necessary.
- This is subjective, but I'd be very interested in a more thorough analysis of advantages and disadvantages of reusing representations for learning (in other words, what is the impact of having an arrow between the u-components in Figure 1). From my perspective, this is the main difference between the "mainstream" lifelong learning approaches and the proposed method and it should be investigated more thoroughly.
- The paper should address the comments about related work and minor issues listed above.

**Strengths And Weaknesses:**

Strengths: - The work sets an ambitious goal of achieving positive backward transfer rather than no forgetting. This is probably one of the most interesting problems in continual learning now.
- The approach is simple, intuitive and easy to understand. In particular I think it could be used as a strong baseline for lifelong learning approaches working with modularity. As such, I think the paper will be interesting for the TMLR audience.

Weaknesses:
- I'm not convinced by the evaluation protocol. Rather than use metrics existing in other continual learning works, the authors introduce their own set of evaluation criteria. The authors claim that the metrics used in [1] are not sufficient. But even if that's the case, there are several works that introduce different metrics, especially for measuring transfer [2, 3, 4]. The authors point out that their transfer metrics have some nice properties, e.g. they can be decomposed into task-specific transfer or forward and backward transfer, but I believe this should be achievable with other metrics as well, as they are often just sums (or averages) of per-task transfers. Finally, the authors also argue that the metric should report relative gains in performance rather than absolute. I personally do not agree. Obviously, they have certain advantages, but there are also drawbacks -- relative metrics might be particularly noisy when the performance is very high (e.g. the difference of 99.9% vs 99.8% accuracy might often be due to noise, but the relative metrics will amplify this gap). I raise this point as: (1) there are already many existing metrics in CL literature which makes direct comparison more difficult and I believe we should go towards unifying rather than branching out, and (2) in the experimental section there are some experiments where the proposed method seems to be worse from the perspective of the "traditional" metrics but better from the perspective of the introduced metrics which makes it possible that metrics were chosen in order to fit the method well.
- The methods used here for baselines are far from SOTA. In particular, the main point of comparison for SynN seems to be Progressive Networks [5] which builds up representations for different. Since then, numerous methods building on this premise were proposed [2, 6, 7] and some of them even support the resource-constrained flavor of CL. I do not necessarily expect a direct comparison with all these methods as it might be prohibitively expensive computationally, but I think they should be acknowledged and discussed. Additionally, I think it would be safer not to refer to the tested baselines as state-of-the-art methods, as stated e.g. in Section 5.3.2.
- The previous point is especially important, as I think the modularity-based methods are the most direct "competitor" of the proposed approach in deep learning. They enable more efficient learning of representations for the new tasks by reusing the representations for the old task. This is in contrast with SynN which learns representations separately for each task. The authors describe this as "benefiting from past representations without biasing future representations" which I think is overly pessimistic, especially in realistic scenarios where tasks are not created adversarially. I think there is a crucial question here that haven't been explicitly asked: "What do we lose or gain if we do not reuse representations from previous tasks for learning the new task?". This criticism is a bit subjective as one might consider different questions to be important. However, given the current trends in transfer learning, I think it is extremely relevant and a more thorough comparison with modularity-based CL methods would be welcome.
- The authors claim that the Model Zoo method build on their approach, but looking at the Model Zoo paper I don't see this clearly stated. I think the paper should further clarify this issue.
- The paper focuses on the task-aware setting of lifelong learning that is currently considered to be a fairly easy setting. In particular, most replay-based papers focus on the incremental class learning scenario where the task ID is not given during evaluation.
- I think related work concerning other approaches to ensembling for lifelong learning should be extended. In particular, Section "Ensemble Methods" in [8] lists methods such as Learn++, TrAdaBoost and Accuracy Weighted Ensembles that are not discussed here.

Minor issues:
- Abstract: "classical machine learning, which we define as, starts from a blank slate" -> part of sentence missing
- Section 2.2 "constrains for lifelong learning is crucial" -> "are crucial"
- Section 2.3 "number of episode" -> "number of episodes"
- Section 4 "both bagging and boosting asymptotically converges" -> "converge"
- Section 5.3 "see Appendix Figure 2" -> the reference here goes to figure 2 in the main text.

[1] Díaz-Rodríguez et al., Don't forget, there is more than forgetting: new metrics for Continual Learning, arXiv 2018 \
[2] Veniat et al., Efficient Continual Learning with Modular Networks and Task-Driven Priors, ICLR 2021 \
[3] Bornschein et al, NEVIS'22: A Stream of 100 Tasks Sampled from 30 Years of Computer Vision Research, arXiv 2022 \
[4] Hayes et al., New Metrics and Experimental Paradigms for Continual Learning, CVPR 2018 workshops \
[5] Rusu et al, Progressive Neural Networks, arXiv 2016 \
[6] Allya et al., PackNet: Adding Multiple Tasks to a Single Network by Iterative Pruning, CVPR 2018 \
[7] Ostapenko et al., Continual Learning via Local Module Composition, NeurIPS 2021 \
[8] Kemker et al, Measuring Catastrophic Forgetting in Neural Networks, AAAI 2018

---

> ### Author Response · Authors · 2023-03-20
> **Thanks a lot for providing us with your thoughtful feedbacks and valuable time!**
>
> We heartily thank reviewer szZq for the nice suggestions and specially the references which have enriched our knowledge and made our paper better! We tried to address the concerns in the following few lines:
>
> - "I'm not convinced by the evaluation....."
>
> Reply:
>
>  . Changes made: Section 3.2, Figure 10 and Appendix Table 10.
>
>  . Summary:  We discussed our weaknesses in the last paragraph of Section 3.2 as suggested by the reviewer. We agree with the reviewer’s concern about unifying the performance metrics for lifelong learning. However, we also believe that there is no single metric that dominates all other metrics in all scenarios. Our goal is to show that some properties are desirable for these metrics so that models with different levels of accuracy and hyper-parameters can be compared based on only how well they can transfer knowledge between different tasks. In this vein, we demonstrated that our proposed metrics have some of these desirable properties. We have also tested the benchmarking algorithms using the metrics developed by Veniat et al.
>
> - "The methods used here for baselines are ..."
>
> Reply:
>
>  . Changes made: Removed SOTA word, Section 1 (paragraph 4,5,6)
>
>  . Summary: We have discussed the suggested methods along with their relative advantages and disadvantages.
>
> - "The previous point is especially important, as ..."
>
> Reply:
>
>  . Changes made: Section 7 (Paragraph 1)
>
>  . Summary: We thank the reviewer for pointing out an interesting idea! However, we are unsure of the effect of learning new encoders dependent on the past ones and thereby, bagging all the encoders using a channel layer. In this case, the subsequent encoders will be more correlated to the past ones. This is unclear whether ensembling the dependent encoders will or will not reduce the variance of the estimated posteriors compared to those of our proposed approach while doing forward learning. Moreover, learning lateral connections between the encoders will render our approach computationally inefficient like ProgNN as shown in Appendix E.3. In this paper, we tried to establish bagging of encoders as an effective ensembling approach for lifelong learning. Introducing new concepts may deviate  the readers from the main message of our work. Nevertheless, the proposed study is an interesting question to explore. Therefore, we have mentioned the proposed study as our future work in Section 7. We agree that unlike the modular methods, our approach learns a new encoder for each new task. This is not an optimal way to grow the capacity of the model. We tried to solve the capacity growth issue differently. After we have learned a large number of encoders, the incremental benefit from training new encoders becomes negligible. For example, we have added a new experiment on a large scale dataset in Figure 9 where the BLE curves eventually saturate. We have also highlighted this point in the introduction (last 2 lines) and the resource recycling experiment in Section 6.1.3.
>
> - "The authors claim that the Model Zoo ..."
>
> Reply: For the sake of anonymity, we are unable to disclose the details of the claim. We are friends and collaborators with the authors of Model Zoo, and we discussed prior pre-print versions of our manuscript with them many times. Boosting is the straightforward next step after bagging. However, after going through Learn++, TrAdaBoost as suggested by the reviewer, we realized the intuition behind Model Zoo is not novel and boosting of learners for continual learning has been used in the past. We have removed our claim.
>
> - "The paper focuses on the task-aware setting ..."
>
> Reply: In this paper, we have mainly focused on proposing the possibility of backward transfer using the bagging approach of ensembling encoders. To simply describe the idea, we have used the simplest possible settings. We have acknowledged the limitations of the work in the first paragraph of Discussion (Section 7). We will pursue the more challenging task unaware setting in our future work.
>
> - "I think related work concerning other approaches to ensembling ..."
>
>  . Changes made: Section 1 (paragraph 4)
>
>  . Summary: We have discussed the suggested methods.
>
> We have fixed all the minor issues.

---

### Review · Reviewer_8Tcb · 2023-02-01

**Summary Of Contributions:**

The paper proposes "representation ensembling" for lifelong learning. The idea is to create a new encoder for every new task and then ensemble the representations to improve forward/backward transfer. Based on this idea, two variants are proposed, one for neural networks and the other for random forests. Further analysis in the paper reveals the complexity of the proposed algorithm achieves linear complexity.

**Audience:**

Yes

**Broader Impact Concerns:**

No concerns.

**Claims And Evidence:**

No

**Requested Changes:**

1. I suggest rewriting section 4 in a more clear way. For example, write the schedules about what to do when a new task joins. How does the architecture changes and what kind of data is used.

2. Address my questions related to the complexity analysis.

3. Adding more realistic benchmarks. Current benchmarks are small-scale. Redo the figures to clarify the differences among different algorithms.

**Strengths And Weaknesses:**

# Strength
1. The idea is valid. It is a natural extension of ensembling the decisions.
2. The analysis of algorithm complexity is interesting.
3. The experimental validation is performed on a variety of benchmarks.

# Weakness
1. The approach description is unclear and confusing, although the idea is not that hard to understand. There are some details that need to be addressed. For example, how does the proposed algorithm differ from a mixture of experts? I suggest a rewriting of the methodology part.
2. The complexity analysis is a bit confusing as well. Intuitively, training time is related to the number of epochs so I think the time complexity is a factor of n. And this epoch number can be pretty large which leads to significantly different performances. I wonder what the authors think about this question.
3. Although the experiments seem to have various benchmarks, these benchmarks are small scale and it is hard to draw any concrete conclusions from these experiments. In addition, the curves are tangled and hard to understand. The colors for different algorithms are hard to distinguish.

---

> ### Author Response · Authors · 2023-03-20
> **Thanks a lot for proving us with valuable feedbacks and making our work better!**
>
> We are grateful to reviewer 8Tcb for precisely pointing out our weaknesses and help us learn more about lifelong learning! We tried to address the concerns in the following few lines:
>
>  - "I suggest rewriting section 4 in a more ..."
>
> Reply:
>
>  . Changes made: Section 4, Appendix C.
>
>  . Summary: We have rewritten section 4 and described the approach along with the pseudocodes in Appendix C.
>
>  - "Address my questions related to the complexity analysis"
>
> Reply:
>
>  . Changes made: Section 4.3, Appendix E.
>
>  . Summary: We have added our assumptions in Section 4.3 and showed the time complexity analysis in Appendix E.
>
> - "Adding more realistic benchmarks. Current benchmarks ..."
>
> Reply:
>
>  . Changes made: Section 6.4, Figure 3, 7, 8, 9, Appendix Figure 3, 4.
>
>  . Summary: We have added a new benchmark on a large-scale FOOD1K dataset with 50 tasks in Section 6.4. But because there was a huge computational load for the experiment, we only showed the results of the top performing four algorithms in other experiments. We have also modified all the figures to clarify the differences among different algorithms.

---

### Review · Reviewer_K9wf · 2023-02-27

**Summary Of Contributions:**

The paper proposes an algorithm for continual learning that systematically enables forward and backward transfer. The algorithm is based on ensembling representations in quasilinear space and time. The paper demonstrates instantiations of the proposed algorithm based on decision trees and neural networks, and shows that it performs competitively in enabling transfer in continual learning. Further, it introduces three metrics to benchmark transfer in a continual data stream.

**Audience:**

Yes

**Broader Impact Concerns:**

None in particular

**Claims And Evidence:**

No

**Requested Changes:**

Other suggested corrections (beyond the weaknesses above):

* Pages 18 and 19 can be moved to the appendix. If some plots are also removed and the paper is restructured, the paper would be more readable.
* Regarding algorithms 1,2,3 and 4 in the appendix: I would suggest the authors describe what is done by each algorithm in a few words, and describe the order of algorithms to be used when training or inference.
* Figure 7 - second plot on the top row is not understandable as there are too many lines. I suggest the authors take note of this and remake the plot.
* In the Benchmark Data Experiments (subsection 5.3) → each dataset should be a subsection, and you can evaluate all algorithms under each sub-subsection. Sub-subsections 2, 3, 4 and 5 are not on different datasets but seem like ablation studies; it is also not clear which dataset this has been conducted on, as the plots/figures (such as Figure 3 and Figure 6). I would suggest the authors use tables for the large-scale experiments of CIFAR100 and MiniImageNet.
* I suggest the authors reorganize subsubsection 5.3.8 and possibly move the entire experiment to the appendix (since it is just a verification that your approach works on that domain, and is not a benchmark dataset).
* I suggest the authors to move Figure 8 to the appendix as it is a dataset detail and does not relate to the paper.
In the introduction, the authors state: “Moreover, when we desire to transfer backward, we keep the prior data, like replay approaches”. I would suggest rewording this sentence. It is misguide a reader towards the interpretation that using replay is the only possible way to achieve backward transfer.
* In section 2.2, before equation 3, the authors state that a “weight” can be assigned to each task depending on its importance/priority. However, this is not the goal of continual learning at all, and it is unclear why the authors have interpreted continual learning in this way in this section (although they use the right interpretation in their algorithm).
* In the first line of page 4, the authors state “both store all the data they have ever seen”, which is not true: partial replay operators under a memory constraint that only k (k << N) examples out of the total N examples of a task can be stored.
* In section 4, the authors state “Next, by pushing the held-out dataset through the tree, we can learn the channel, i.e., posteriors in the leaf-nodes.” It is unclear where the held-out dataset comes from why this is needed to learn the channels. The held-out dataset usually refers to the validation set so it is better to reword the sentence if the authors mean something else.
* I would suggest adding citations to algorithms again in section 5, for improved readability.


**Strengths And Weaknesses:**

Strengths
=========
+ The paper introduces new metrics to measure transfer. The paper explains the choice of metrics intuitively. These metrics will be useful to benchmark transfer in the continual learning setup.
+ The paper attempts to benchmark the computational complexity of algorithms with respect to sample size - this is a laudable effort as such benchmarking is absent in most existing papers, specifically in the architecture-growing setup in continual learning.
+ The paper introduces interesting toy datasets of continual learning streams to measure forward and backward transfer
+ The paper has presented the proposed algorithm in a highly generalized way, in the context of having an encoder, channel and decoder. Although the terms used may be questionable, this is useful as it allows them to bring in simple models such as decision trees in their toy examples and to intuitively explain the complexity.

Weaknesses
==========
- The algorithm operates in a fairly unconstrained setup, and there is no high-level explanation of why the algorithm will scale:
   * The algorithm uses a single encoder for each task which is not possible to scale unless there are constraints on how large each encoder can be - and it is non-trivial to understand from the first reading of the paper why such a methodology is useful and why it will scale. The one difference the proposed algorithm has compared to ProgNN is that it does not have lateral connections between the encoders. However, the proposed algorithm operates in a fairly unconstrained setup of continual learning: it expands with a single encoder per task, and also stores data. Can you please explain in plain words in the introduction why your approach scales compared to a naive learner that uses a single network per task, before you get into the details?

- Unclear setting of operation in continual learning:
   * Are the task identities known at test time?
   * Does the algorithm use the single-pass (online) setup for each task or the multi-pass (offline) setup?
   * How are exemplars chosen from each task?

- This paper has not been presented well, I had a difficult time navigating the paper to place and evaluate the algorithm in context. I would suggest the authors kindly take note of some of the points below to restructure their paper:
   * It is not understandable which metric has been used when referring to “accuracy” and “single-task accuracy” (is it the averaged incremental accuracy measured at the end of all tasks, or averaged at each task in the stream, or is it something else?). I suggest the authors include the formulate used to compute the (averaged) accuracy and single-task accuracy in the paper.
   * There are no numbers in the entire paper and the results are completely in the form of plots. Unfortunately the plots are also too small and it is difficult to understand and evaluate the algorithm. I would suggest using tables instead (all metrics together, or one table per metric), and removing or moving “single-task learners” to the appendix. I would otherwise suggest adding all methods within the same large plot and use different markers for resource-constrained and resource-unconstrained methods. Again, since there is a distinction between these methods, it is best to use a table and indicate it with the specific complexities (either in the table or in the table’s description).
   * What is the difference between capacity and space complexity?
   * "Thus,an algorithm that literally stores all the data it has ever seen, and retrains a fixed size network on all those data with the arrival of each new task, would have smaller space complexity and the same time complexity as ProgNN.” - it is unclear how the time complexity for ProgNN also scales quadratically with n when n is proportional to T. Do the authors refer to offline training? In that case, don’t SynNN and SynF also have the same time complexity, thereby suffering from the same limitation?

- Insufficient experiments to understand/evaluate the algorithm:
   * There are no upper bounds in the experiments  - training a single network  with all the data, with the same capacity as that of SynNN after all tasks, to understand how close the framework gets to the “best possible value”, and also to understand the importance of the introduced framework in the context of a learner that sees all data IID and has the same total network capacity.
    * The paper has used exemplars in its algorithm but has not denoted the exemplar size in any of its experiments (which is important to know, as the framework uses exemplars, but also expands in capacity). The paper also needs to benchmark its algorithms with varying exemplar memory sizes.
    * No ablation experiments: the paper needs to put the proposed algorithm apart and measure the relative contribution of each component. It is unclear if storing the  data helps or growing the model’s architecture helps.
    * It is unclear what single-task learners do, and why they do not reach the best possible value for the continual stream - I expected that training a single network for each task would be optimal in terms of the accuracy - can you elaborate why they reach the same value as a continual learner?
    * The paper proposes two versions of the algorithm: non-parametric and semi-parametric. But it is unclear which version has been used in experiments throughout the paper.

- Insufficient comparison to similar algorithms:
    * [1] also lies in the same regime as the paper: it expands representations, and also stores exemplars. The paper should benchmark this algorithm (the code is openly available) and compare it with the proposed one, across all metrics of accuracy and transfer.
    * The paper states in page 10 that “SynF is the only non-parametric lifelong learning algorithm to our knowledge.”. I’d like to point out that that is not true, there is [2] which is also non-parametric. The paper should compare its algorithm to [2], both in the computational complexity table as well as in the results.

- Insufficient information for reproducibility:
    * In the appendix, table 1 reports hyperparameters for SynF used in CIFAR. Is this for CIFAR-10 or CIFAR-100? Where are the hyperparameters for the other datasets? Also, where are the hyperparameters for SynNN?
    * Are the results from other papers reported directly from the papers, or have they been reproduced? In case the results have been reproduced, did the paper use the same hyperparameters used in the original papers? Did the paper ensure that the network size (number of parameters) are fairly the same across all the papers or they are comparable? Has the same number of exemplars been used in all the papers? All of these questions are important for the results to be comparable and reproducible.

References:

[1] DER: Dynamically Expandable Representation for Class Incremental Learning, CVPR 2021, https://openaccess.thecvf.com/content/CVPR2021/html/Yan_DER_Dynamically_Expandable_Representation_for_Class_Incremental_Learning_CVPR_2021_paper.html

[2] Continual Learning using a Bayesian Nonparametric Dictionary of Weight Factors, AISTATS 2021, https://arxiv.org/abs/2004.10098

---

> ### Author Response · Authors · 2023-03-20
> **Thanks a lot for providing us with thoughtful feedbacks and lending us your valuable time!**
>
> We thank reviewer K9wf for the nice and detailed feedbacks! We learned a lot about lifelong learning methods while trying to address the concerns! We tried to address the concerns in the following few lines:
>
>  -  "The algorithm uses a single encoder for each task .. "
>
> Reply:
>
>  . Changes made: Section 6.1.4.
>
>  . Summary: We have two implementations for our approach: SynF and SynN. SynF uses decision forest as encoders for each task. As decision trees are non-parametric models, they can expand according to the complexity of the task. However, for SynN we assume equal complexity for each task while training the encoders. Note that even if the complexity of the new task is higher compared to the capacity of each encoder, our approach can leverage other task encoders through the channel layer to fit the current task. We solve the scaling issue slightly differently which is mentioned in the last two lines of introduction. We have two stages as shown in the recycling experiment on CIFAR data (Figure 3 bottom right): resource building and resource recycling. Once we build enough resources from the past tasks, we do not need to train new encoders for the new tasks. We can recruit the old ones to perform optimally on the current tasks. After we have learned a large number of encoders, the incremental benefit from training new encoders becomes negligible. For example, we have added a new experiment on a large scale dataset in Figure 9 where the BLE curves eventually saturate. Meanwhile, we hope to pursue resource recycling further in our future work as mentioned in the experiment. Moreover, we reorganized the single task expert experiment in Section 6.1.4 which shows our approach can significantly do better than a naive learner that uses a single network per task.
>  - "Unclear setting of operation in continual learning.."
>
> Reply:
>
> . Changes made: Section 1 (last paragraph lines 6-10)
>
> . Summary: We have clarified the setting and exemplar size.
>
>  - "It is not understandable which metric has been used when referring to “accuracy” and “single-task accuracy” .."
>
> Reply:
>
>  . Changes made: Section 6.1.4.
>
>  . Summary: We have clarified the experiment and used formula to define average accuracy.
>
>  - "There are no numbers in the entire paper ..."
>
> Reply:
>
>  . Changes made: Figure 3, 5, 7, 8, 9, Appendix Figure 3, 4, Appendix Table 4, 5, 6.
>
>  . Summary: We have redone the Figures and added tables with numerical performance measures.
>
> - "What is the difference between capacity..."
>
> Reply:
>
>  . Changes made: Section 4.3 (first paragraph).
>
>  . Summary: We have defined capacity and space complexity.
>
>  - "Thus,an algorithm that literally stores all the data it has ever ..."
>
> Reply:
>
>  . Changes made: Section 4.3 (last 2 paragraph), Appendix Section E.
>
>  . Summary: SynN and SynF has linear time complexity as the time required for pushing the old task data though the old encoders and learning or updating channels is negligible in comparison with the time required for training a new encoder. We have derived the time complexity for ProgNN in Appendix E.
>
>  - "There are no upper bounds in the experiments..."
>
> Reply: We have trained a random forest with the same number of trees (100 trees) as SynF would have after being trained on all the 10 tasks in CIFAR 10X10 (500 samples per task). For SynN, we have trained a deep net model with the number of convolutional channels 10 times as that of a single column of SynN. Then we trained both models  on the whole dataset for 10 tasks. We achieved 14.15% accuracy for random forest and 15.13% accuracy for the deep net model. However, as shown in Figure 4, SynN and SynF can reach more than 40% accuracy after having seen all the tasks. This suggests training a single network with all the data, with the same capacity as that of SynF or SynN after all tasks, cannot be an upper bound for our proposed approaches. This phenomenon of relative advantages between doing isolated (10 classes in each isolated training) and joint training of the classes (total 100 classes) in the dataset is well-studied in the literature [1], [2]. Training in isolation amounts to approximating the non-linear decision boundaries between different classes with piecewise linear decision boundaries which are easier to learn compared to that of the joint training.
>
> [1] Friedman, Jerome H. "Another approach to polychotomous classification." Technical Report, Statistics Department, Stanford University (1996).
>
> [2] Hastie, Trevor, and Robert Tibshirani. "Classification by pairwise coupling." Advances in neural information processing systems 10 (1997).

---

### Decision · Action_Editors · 2023-05-08

**Recommendation:** Reject

**Comment:**

The reviewers all found the work to be of interest to the TMLR audience and found the problem setting interesting. Several minor comments raised by the reviewers were addressed by the authors in the revision (ie several Figures were updated). However, there remained significant concerns about the experimental validation. Salient points raised by the reviewers were:
- A more recent baseline than progressive networks should be used. If this is not possible, a more thorough discussion of literature and a clear statement of limitations is necessary.
- The analysis and ablation studies continues to be limited; the paper can benefit from more deeper analysis of design choices and hyperparameters.
- The papers lacks comparison with or even sufficient discussion of modularity-based CL methods which seem to be very relevant.

Addressing these comments will require changes that warrant another round of review. I encourage the authors to resubmit after making these improvements.

**Audience:**

This work is relevant to the TMLR audience.

**Claims And Evidence:**

This work proposes a method for lifelong learning based on ensembling representations from different tasks. The algorithm is proposed in a general way such that it can be applied to different methods (neural nets and decision trees) and an analysis of the algorithm complexity is provided. The experimental validation is performed on a range of benchmarks and it is shown the algorithm performs competitively in enabling positive transfer in continual learning.